# Genome editing reveals fitness effects of a gene for sexual dichromatism in Sulawesian fishes

Satoshi Ansai 1,2,9, Koji Mochida3,4, Shingo Fujimoto3,10, Daniel F. Mokodongan3,11,
Bayu Kreshna Adhitya Sumarto3, Kawilarang W. A. Masengi5, Renny K. Hadiaty6,12, Atsushi J. Nagano 7,
Atsushi Toyoda 8, Kiyoshi Naruse 2, Kazunori Yamahira3 & Jun Kitano 1✉

Sexual selection drives rapid phenotypic diversification of mating traits. However, we know little about the causative genes underlying divergence in sexually selected traits. Here, we investigate the genetic basis of male mating trait diversification in the medaka fishes (genus *Oryzias*) from Sulawesi, Indonesia. Using linkage mapping, transcriptome analysis, and genome editing, we identify *csf1* as a causative gene for red pectoral fins that are unique to male *Oryzias woworae*. A *cis*-regulatory mutation enables androgen-induced expression of *csf1* in male fins. *csf1*-knockout males have reduced red coloration and require longer for mating, suggesting that coloration can contribute to male reproductive success. Contrary to expectations, non-red males are more attractive to a predatory fish than are red males. Our results demonstrate that integrating genomics with genome editing enables us to identify causative genes underlying sexually selected traits and provides a new avenue for testing theories of sexual selection.

[1] Ecological Genetics Laboratory, Department of Genomics and Evolutionary Biology, National Institute of Genetics, Mishima, Shizuoka, Japan. [2] Laboratory of Bioresources, National Institute for Basic Biology, Okazaki, Aichi, Japan. [3] Tropical Biosphere Research Center, University of the Ryukyus, Nishihara, Okinawa, Japan. [4] Department of Biology, Keio University, Yokohama, Kanagawa, Japan. [5] Faculty of Fisheries and Marine Science, Sam Ratulangi University, Manado, Indonesia. [6] Research Center for Biology, Indonesian Institute of Science (LIPI), Cibinong, Indonesia. [7] Faculty of Agriculture, Ryukoku University, Ohtsu, Shiga, Japan. [8] Comparative Genomics Laboratory, National Institute of Genetics, Mishima, Shizuoka, Japan. [9] Present address: Graduate School of Life Sciences, Tohoku University, Sendai, Miyagi, Japan. [10] Present address: Department of Human Biology and Anatomy, Graduate School of Medicine, University of the Ryukyus, Nishihara, Okinawa, Japan. [11] Present address: Museum Zoologicum Bogoriense (MZB), Zoology Division of Research Center for Biology, Indonesian Institute of Science (LIPI), Cibinong, Indonesia. [12] Deceased: Renny K. Hadiaty. ✉email: jkitano@nig.ac.jp

Sexual selection is an important evolutionary force that can drive rapid phenotypic diversification within and among species[1–5] and even promote speciation[6,7]. For example, competition for mates drives the evolution of traits that can increase attractiveness to the opposite sex and/or increase competitiveness with the same sex[1]. These sexually selected traits, however, can decrease the survival rates of possessors, for example, by increasing the probability of detection by predators[8–13]. Therefore, sexually selected traits are predicted to be expressed only in the sex for whom the trait is beneficial, which is generally the sex with higher variance in mating success[14,15]. Indeed, there are many examples in which sexually selected traits, such as ornaments and armaments, are sex-specific[1,2].

Despite its importance, there are few studies that have identified genes underlying divergence in sexually selected traits between species, particularly in vertebrates[16]. Identification of the causative genes enables us to answer several important questions related to sexual selection. First, it will help to understand the genetic mechanisms by which only one sex expresses a sexually selected trait (i.e., resolution of intra-locus sexual conflict[17–19]). Previous studies have shown that linkage to sex chromosomes and regulatory mutations, such as changes in gene expression by androgen signaling in vertebrates, are two main mechanisms to break the genetic correlations between males and females[20–25]. By identifying the causative genes, we can directly answer whether these or other mechanisms are involved.

Second, genetic manipulation of the causative gene enables us to directly test how particular genetic changes affect each fitness component in males and females. The majority of previous studies have examined phenotype-fitness correlations in the field and/or in the laboratory, or physically manipulated the phenotypes (e.g., cutting or attaching the ornament)[2]. Although these studies have demonstrated the fitness effects of a phenotype, we know little about the fitness effects of causative genes underlying that phenotype. Many genes are pleiotropic and have multiple functions; therefore, identifying the functional roles of causative genes in multiple fitness-related phenotypes is essential to fully understand the evolutionary forces and constraints acting on the causative gene. Recent advances in targeted genome-editing technologies with the CRISPR/Cas system makes it possible to manipulate genes in vivo relatively easily and investigate the fitness effects of a gene. To our knowledge, however, there has been no study using this technology to investigate the fitness effects of causative genes underlying sexually selected traits.

The family Adrianichthyidae, commonly referred to as medaka, consists of two genera and 37 species (genus *Adrianichthys*: four species; genus *Oryzias* 33 species) of teleost fishes[26–28]. Although they are broadly distributed throughout East and Southeast Asia, 20 of these species are endemic to Sulawesi, the largest island in Wallacea[26–28] (Fig. 1a). Interestingly, endemic species in Sulawesi exhibit considerable diversification in sexually dimorphic traits, such as the shape and color of body and fins[29]. Because of the ease of breeding these species in laboratory conditions and the availability of genome-editing techniques[30], they have the potential to be a good model system to explore the molecular mechanisms underlying the evolution of sexually selected traits.

In this study, we first sequence and assemble the genome of *Oryzias celebensis* to make a reference genome sequence for the Sulawesian medaka fishes. Next, we investigate the molecular genetic basis of sexual dichromatism in *O. woworae*. By quantitative trait locus (QTL) mapping, comparative transcriptome analysis, and genome editing, we identify an autosomal gene *csf1* as a causative gene for the red coloration in the pectoral fins. Finally, using genetically engineered *O. woworae* with targeted disruption of *csf1*, we investigate the functions of this gene in survival, female mate preference, and attraction to predators.

## Results

### Endemic diversification of medaka fishes in Sulawesi Islands.
To establish the Sulawesian medakas as a model for genetic studies of phenotypic diversification of male mating traits, we first made a reference assembly of *O. celebensis* using a female fish with XX sex chromosomes maintained as a closed colony in the laboratory for ~40 years. Using a combination of PacBio sequencing, high-coverage Illumina short-read sequencing, and optical mapping with the Irys system, we made a reference assembly containing 688 Mb (631 scaffolds). Among them, 57 scaffolds were anchored to 18 linkage groups (LGs) constructed by a linkage map of a $F_2$ family of *O. celebensis* × *O. woworae*. This LG number (18) is equivalent to the previously reported chromosome number of *O. celebensis*[31]. This assembly contained 24,120 protein-coding genes identified by RNA-seq of multiple tissues. Comparison with the publicly available genome sequence of the Japanese medaka (*O. latipes*) showed clear synteny (Supplementary Fig. 1), suggesting that our assembly was successful. We named each chromosome of *O. celebensis* based on the gene synteny with the Japanese medaka. This final assembly was used as a reference for the following analysis unless noted.

Next, we resolved the phylogenetic relationships of Sulawesian medaka fishes. To this end, we sequenced the whole genomes of 17 endemic species of Adrianichthyidae with Illumina short reads and mapped them to the *O. celebensis* reference. Using the sequences of 10,174 single-copy genes, we constructed a maximum-likelihood phylogenetic tree (Fig. 1b). Our divergence time estimation showed that these 17 species diversified within the last 14.40 million years (95% Credible Interval [CI] = 11.11–18.65) and 15 species of the genus *Oryzias* diversified within the last 4.86 million years (95% CI = 3.51–6.74) within the Sulawesi Islands (Supplementary Fig. 2).

### Identification of candidate genes for the acquisition of red pectoral fins in *O. woworae*.
As the first step toward understanding the genetic mechanisms underlying diversification of sexual dimorphism in Sulawesian medakas, we focused on the evolution of body coloration in *O. woworae*. Males of this species show red coloration in the pectoral fins and blue coloration in the lateral side of the body, whereas in females these colorations are less striking (Fig. 2a)[32]. Blue coloration in the body is shared by *O. asinua* and *O. wolasi*. However, red pectoral fins are unique to *O. woworae* among the Sulawesian medakas (Fig. 2a), suggesting that this trait is likely to be acquired in this species. Using *O. celebensis*, which shows no red coloration in the pectoral fins or blue coloration in the body (Fig. 2a), we conducted QTL mapping. Using 164 $F_2$ hybrids (91 males and 73 females) resulting from a cross between an *O. celebensis* female and an *O. woworae* male, we identified a significant QTL for red coloration of the pectoral fin on LG7 (Fig. 2b; Supplementary Table 2). This QTL explained 16.7% of phenotypic variance. Different quantitative measurements of red coloration consistently identified QTL at overlapping loci (Fig. 2b; Supplementary Table 2). Effect plots showed that alleles from *O. woworae* increased redness of the pectoral fins in males, but the effects were much smaller in females (Fig. 2c, d; Supplementary Fig. 3a, b).

For the lateral body coloration, we identified a significant QTL on LG24 for the relative contributions of the red spectral range (605–700 nm) (Fig. 2b; Supplementary Fig. 3c; Supplementary Table 2). However, QTLs for other measurements of blue coloration were not identified. These results suggest that the blue coloration is polygenic or highly influenced by the environment. Black stripes on the tail fin, which are present in both sexes of *O. celebensis*, but not in *O. woworae*, mapped to LG12_20_13 and LG15 (Fig. 2b), suggesting that all body

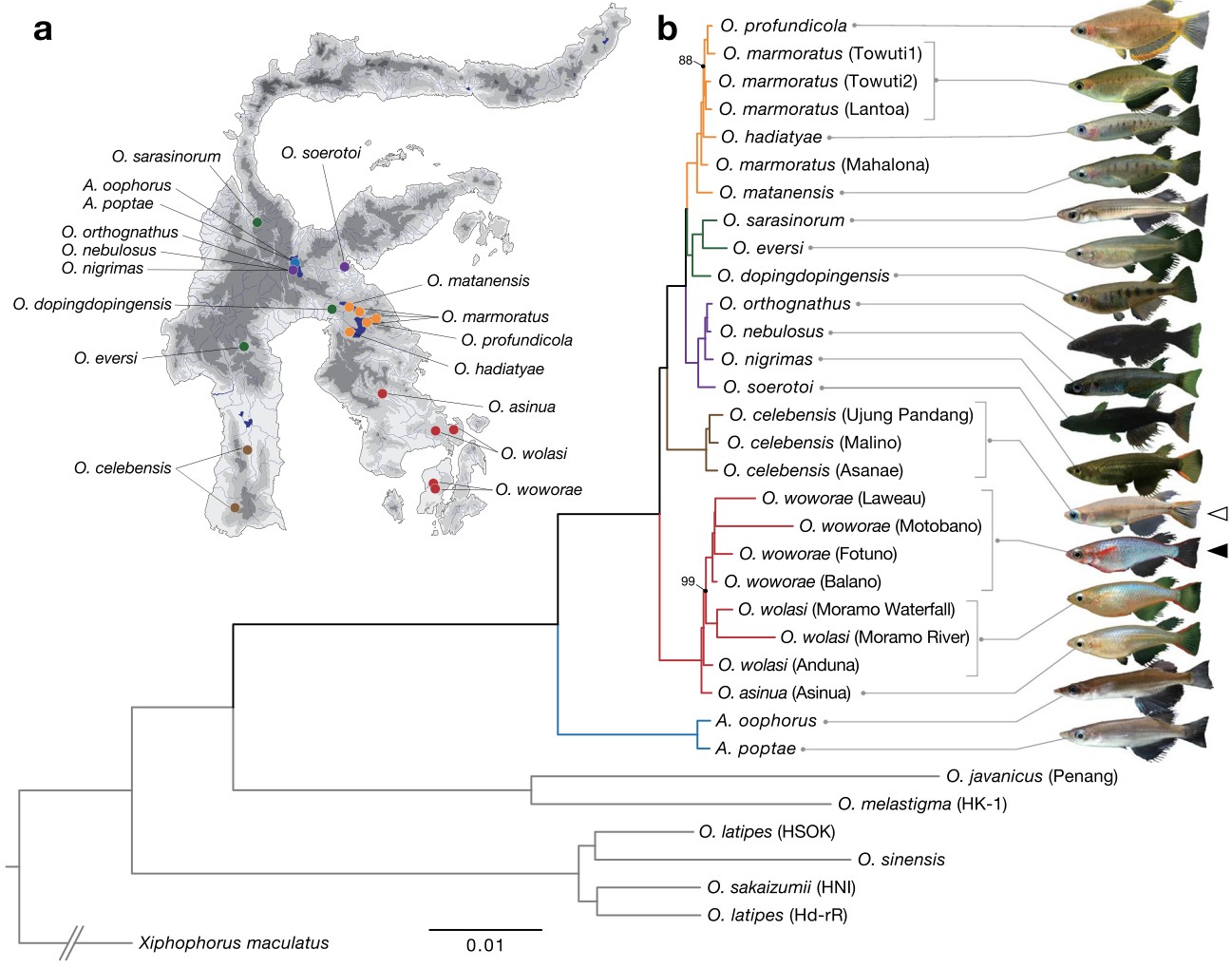

**Fig. 1 Diversification of Adrianichthyidae in Sulawesi. a** A map of Sulawesi showing sampling sites of 17 endemic species. The map was provided by Thomas von Rintelen. **b** A maximum-likelihood phylogenetic tree based on 10,174 single orthologous genes extracted from whole-genome sequencing data. The scale bar indicates the substitution rate. All branches except where noted were supported by 100% maximum-likelihood bootstrap values. Open and closed arrowheads indicate the reference and focal species in this study, respectively. The major lineages of Sulawesi species are shown in different colors.

coloration traits were controlled by different loci. In addition, none of the body coloration traits were linked to sex chromosomes, which are LG24 in both species[33] (Fig. 2b). In addition to the coloration traits, standard length of adult fish differs between the two species, with *O. celebensis* (34.49 ± 5.06 mm for males; 32.79 ± 2.56 mm for females; $n = 6$ each) being larger than *O. woworae* (26.55 ± 0.98 mm for males; females: 26.40 ± 2.58 mm for females; $n = 12$ each) (two-way analysis of variance (ANOVA), species: $F_{1,32} = 53.847$, $P < 0.001$, sex: $F_{1,32} = 0.521$, $P = 0.476$, interaction: $F_{1,32} = 0.632$, $P = 0.432$). QTL mapping of standard length identified a significant QTL on LG12_20_13, again indicating independent control of different traits measured in this study (Fig. 2b; Supplementary Fig. 3d–f; Supplementary Table 2).

Because of the presence of a clear peak for the red pectoral fin QTL on LG7, we searched for candidate genes within the QTL. There were 227 protein-coding genes in the region that contained the 95% CIs of all red fin QTLs. Next, we conducted RNA-seq of the pectoral fins of *O. woworae* males and females and searched for sex-biased genes in the pectoral fins. We also conducted RNA-seq of the pectoral fins of males and females of *O. asinua*, which is more closely related to *O. woworae* than *O. celebensis* but lacks red coloration in the pectoral fins of both sexes (Fig. 1a). We

found 414 and 323 genes expressed differentially between the sexes of *O. woworae* and between males of the two species, respectively (false discovery rate (FDR) < 0.01; Fig. 3a, b). Among these genes, 152 genes overlapped, and only five of them (*csf1*, *magi1a*, *pcbp4*, *ren*, and *slmapb*) were located within the red fin QTL (Fig. 3c). All five genes were highly expressed in *O. woworae* males compared with *O. woworae* females and *O. asinua* (Fig. 3d; Supplementary Fig. 4; Supplementary Table 3).

Among these five genes, one gene encodes the colony-stimulating factor 1 (csf1), which has structural similarities to the Kit ligand[34]. Csf1 signaling is known to be required for xanthophore development in the zebrafish (*Daio rerio*)[34,35]. Although zebrafish has two orthologous copies of *csf1*, *Oryzias* species and other fishes belonging to Acanthopterygii have only one copy of the gene (Supplementary Fig. 5). RNA-seq of the pectoral fins showed that the *csf1* expression level was 3.52 and 9.15 times higher in *O. woworae* males than in *O. woworae* females and *O. asinua* males, respectively (Fig. 3d). We further investigated the expression levels of *csf1* in not only pectoral, but also other fins of *O. woworae* and *O. asinua* males by quantitative real-time PCR. The *csf1* transcript was highly expressed only in fins that show red coloration (pelvic fin and the dorsal and ventral edges of the caudal fin of both species and pectoral fin of

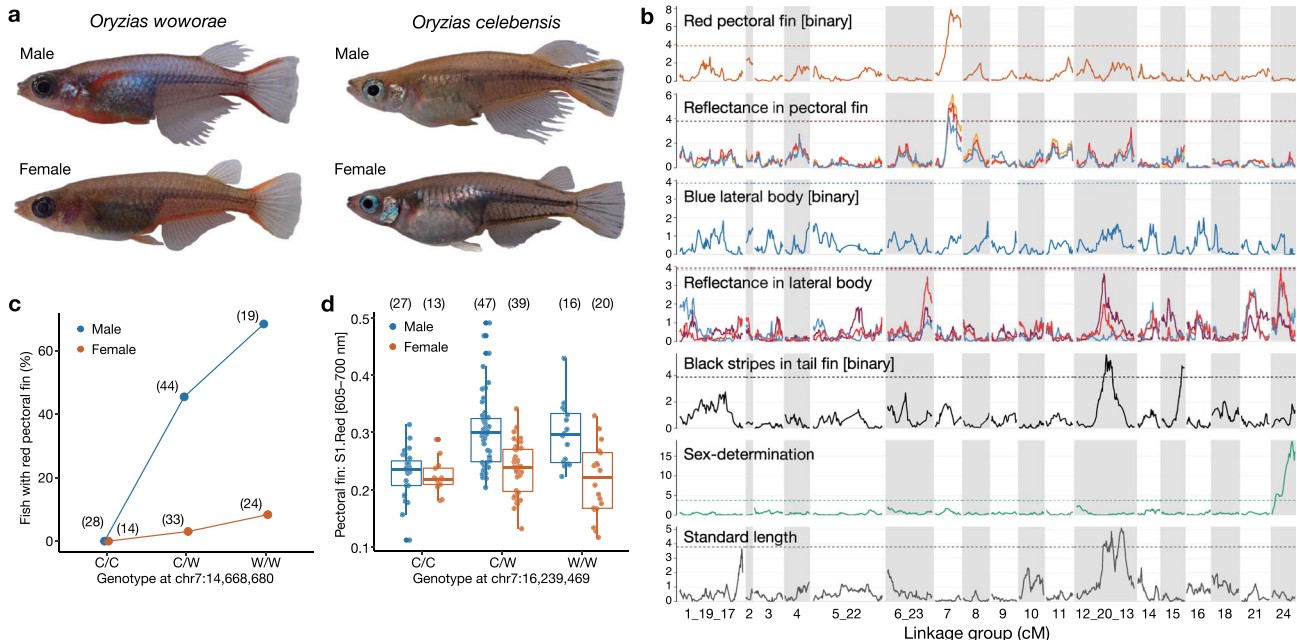

**Fig. 2 Quantitative trait locus (QTL) mapping of body color divergence between _O. woworae_ and _O. celebensis_. a** Representative images of adult males and females of the parental species. **b** Logarithm of the odds (LOD) scores plotted against the chromosomal position on each linkage group. For reflectance values, the different colors indicate the relative contributions of a particular spectral range (red, 605–700 nm; yellow, 550–625 nm; blue, 400–510 nm; violet, 300–415 nm) to the total brightness. The dashed lines indicate the genome-wide significance thresholds determined by 1000 permutation tests ($\alpha = 0.05$). Significant QTLs are summarized in Supplementary Table 2. **c**, **d** Effect plots of the significant QTLs for the red pectoral fins. The _X_ axis indicates the genotypes at each QTL: C/C, homozygous for the _O. celebensis_ allele; W/W, homozygous for the _O. woworae_ allele; C/W, heterozygote. Sample sizes are shown in the parenthesis. The _Y_ axis indicates the percentage of fish with red pectoral fins (**c**) or the contribution of the red spectral range (605–700 nm) to the total brightness (**d**). **d** Each dot represents each individual. The blue and brown colors indicate males and females, respectively. In the box-plots, the center line indicates the median, box limits indicate the upper and lower quartiles, the whiskers indicate 1.5× interquartile range, and the points are outliers. Sample sizes are shown in the parentheses. Source data are provided as a Source Data file.

_O. woworae_) (Fig. 3e). The expression pattern of _csf1_ was further investigated by inserting green fluorescence protein (GFP) into the coding region of _csf1_ using CRISPR/Cas9 in _O. woworae_ (Fig. 3f, g). GFP was expressed where the fins become red (Fig. 3f, g). Thus, _csf1_ is expressed at the right locations and is a good candidate gene for making the pectoral fin red in _O. woworae_.

To directly demonstrate the contribution of _csf1_ to red fin coloration, we next made a _csf1_-knockout (KO) in _O. woworae_ using the CRISPR/Cas system. We isolated a mutant fish with an 11-bp deletion on the first exon of _csf1_ gene (_csf1^{Δ11}_), which causes a frame-shift mutation and is expected to encode a nonfunctional truncated protein (Supplementary Fig. 6). The F_1 heterozygous mutant fish (_csf1^{+/Δ11}_) were further crossed with each other to obtain 163 F_2 adult fish. This F_2 family consisted of 42 wild-type fish (_csf1^{+/+}_), 80 heterozygotes (_csf1^{+/Δ11}_), and 41 homozygous mutants (_csf1^{Δ11/Δ11}_). Because this ratio is close to the Mendelian ratio ($\chi^2$ test, $\chi^2 = 0.0675$, $df = 2$, $P = 0.9668$), this mutation does not substantially influence intrinsic survival. The homozygous mutant males clearly lacked red coloration in any fins, including the pectoral fins (Fig. 4a–d; see Supplementary Fig. 7 for the reflectance spectra data). These data suggest that _csf1_ expression is essential for the development of red fins. Taken together with the expression pattern data, male-specific expression of _csf1_ in the pectoral fin likely underlies the acquisition of red pectoral fins in _O. woworae_ males.

**Increased expression of _csf1_ in pectoral fins by _cis_-regulatory changes**. We first compared the amino-acid sequences encoded by _csf1_ between _O. woworae_ and _O. celebensis_. Although we found four nonsynonymous changes (M163I, S180G, F216L, and

S256N), neither of them was predicted to have a significant effect on the protein functions, based on the PROVEAN algorithm[36] (PROVEAN scores: −0.545, −1.636, 0.560, and −0.645, respectively). This led us to focus on the differences in the expression levels.

To examine whether _cis_-regulatory mutations are responsible for the upregulation of _csf1_ in the pectoral fins of _O. woworae_ males, we compared _csf1_ expression levels among genotypes of the _O. celebensis_ × _O. woworae_ F_2 fish (see above) at the red QTL: homozygous for the _O. celebensis_ allele (C/C), homozygous for the _O. woworae_ allele (W/W), and heterozygous (C/W). Expression levels were significantly higher in W/W and C/W males than in C/C males, with no differences among genotypes in females (two-way ANOVA, genotype: $F_{2,42} = 7.231$, $P = 0.0020$, sex: $F_{1,42} = 8.310$, $P = 0.0062$, interaction: $F_{2,42} = 5.188$, $P = 0.0097$; Fig. 3h; Supplementary Table 4). There were also no differences in expression of an internal control gene (_rpl13a_) among the genotypes (Supplementary Fig. 8). Furthermore, allele-specific expression analysis using the heterozygote showed that the _csf1_ transcript from the _O. woworae_ allele was expressed at higher levels than that from the _O. celebensis_ allele (Fig. 3i; Supplementary Fig. 9), indicating that _cis_-regulatory divergence at the _csf1_ locus causes the increased expression of _csf1_ in pectoral fins of _O. woworae_. Although the _O. woworae_ allele was expressed at a higher level than the _O. celebensis_ allele in both sexes (Fig. 3i), expression levels were much higher in males than in females (Fig. 3h), suggesting that _csf1_ expression is decoupled between the sexes.

As androgen-dependent gene regulation often mediates male-biased expression of autosomal genes in vertebrates[22,23,25], we examined the effects of androgen on _csf1_ expression and red

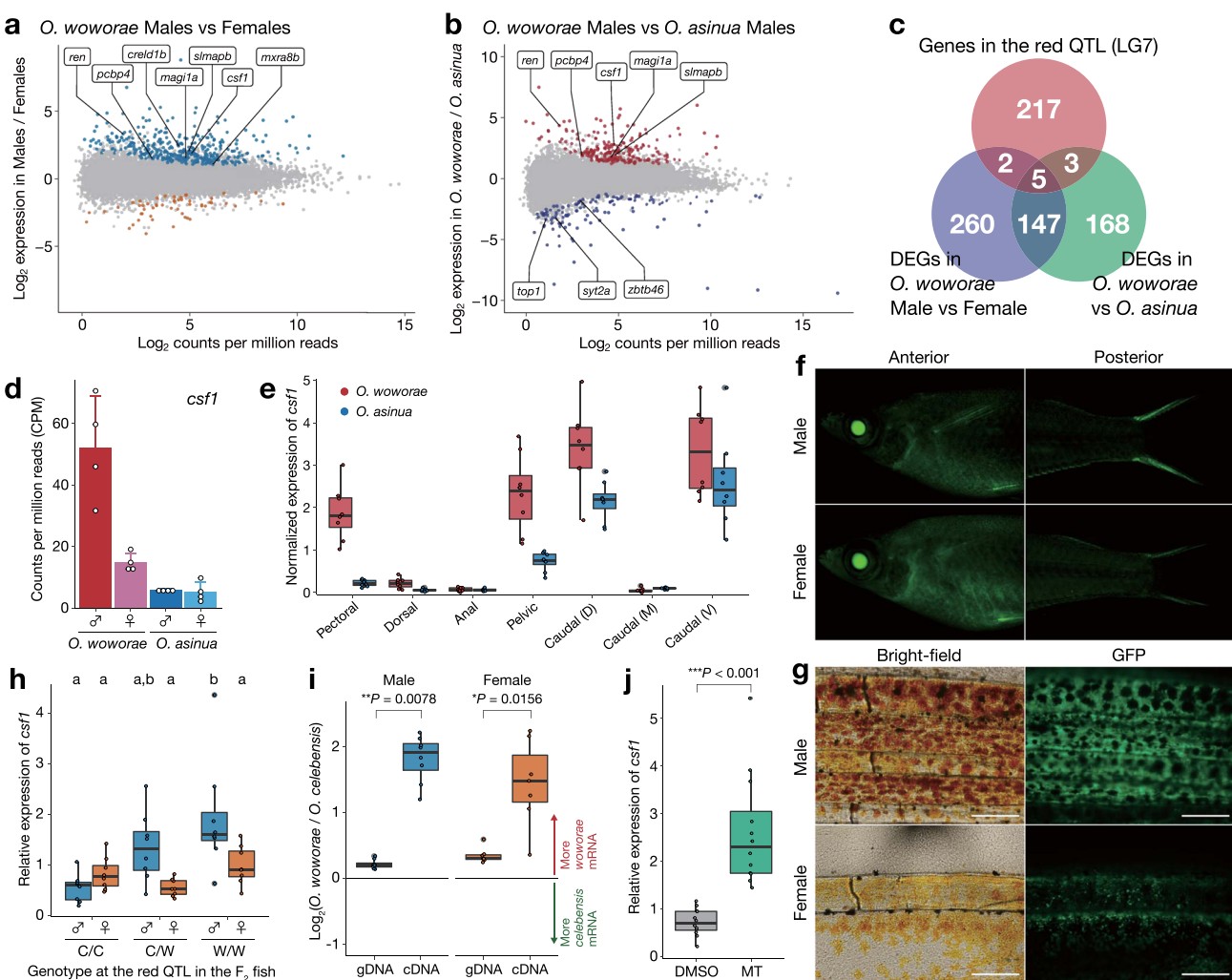

**Fig. 3 Increased expression of *csf1* in male pectoral fins by *cis*-regulatory mutations. a**, **b** RNA-seq of the pectoral fins. Minus-Average (MA) plots with the average gene expression levels on the X axis and log₂ fold-change in expression between the sexes in *O. woworae* (**a**) or between *O. woworae* males and *O. asinua* males (**b**) on the Y axis. Each dot represents a single gene with differentially expressed genes (DEGs) colored (FDR < 0.01). The gene names of DEGs within the red pectoral fin QTLs are also shown. **c** Venn diagram indicating overlap among inter-sex DEGs in *O. woworae*, interspecies DEGs in males, and genes within 95% Bayesian credible intervals of all red pectoral fin QTLs on LG7. Five genes were identified as strong candidate genes. **d** Comparison of *csf1* expression levels among males and females of *O. woworae* and *O. asinua*, showing the highest expression in *O. woworae* males (*n* = 4 for each). Mean ± SD are shown. A dot indicates each individual. Expression levels of the other four candidates are shown in Supplementary Fig. 4. **e** qPCR analysis comparing the relative expression levels of *csf1* in the seven different fin parts of *O. woworae* (red) and *O. asinua* (blue) males (*n* = 8 for each). The expression levels were normalized by the expression of an internal control gene, *rpl13a*. **f** GFP fluorescence in the pectoral (left) and caudal fins (right) in males (upper) and females (lower) of GFP knock-in *O. woworae* (csf1^Olhs:GFP^). Expression in the lens is caused by the basal activity of the heat-shock promoter. **g** Microscopic images of pectoral fins in bright-field (left) and GFP fluorescent observation (right). Images are representative of three experiments. Scale bars indicate 200 μm. **h** Expression levels of *csf1* in the pectoral fin of the F₂ family (*n* = 8 for each). The X axis indicates the genotype at the red pectoral fin QTL on LG7 (C/C, homozygous for the *O. celebensis* allele; W/W, homozygous for the *O. woworae* allele; C/W, heterozygote). Different letters above the boxes indicate significantly different groups (*P* < 0.05, post hoc test with the Tukey's honest significant difference method; see Supplementary Table 4 for the statistical results). **i** Allele-specific expression analysis of *csf1* in pectoral fins of the heterozygous fish (*n* = 8 for each). The ratio of the *O. woworae* allele to the *O. celebensis* allele was significantly higher in cDNA compared with that in genomic DNA. The *P* values are calculated separately for each sex using two-sided Wilcoxon's signed rank test (male, *V* = 36, *P* = 0.007812; female, *V* = 28, *P* = 0.01562; *P* < 0.05 and **P* < 0.01). **h**, **i** the blue and brown colors indicate males and females, respectively. **j** Administration of methyltestosterone (MT) to *O. woworae* females increased *csf1* expression in the pectoral fin compared with the control fish administered dimethyl sulfoxide (DMSO) (*n* = 12 in each group). The *P* value is calculated using two-sided Welch's *t* test (*t*₁₂.₂₃₃ = −5.4148, *P* = 0.00015, ***P* < 0.001). The gray and green colors indicate fish with DMSO and fish with MT, respectively. **e**, **h**–**j** Each circle represents each individual, the center line indicates the median, box limits indicate the upper and lower quartiles, the whiskers indicate 1.5× interquartile range, and the points are outliers. Source data are provided as a Source Data file.

coloration of the pectoral fins. The administration of synthetic androgen methyltestosterone (MT) in *O. woworae* females increased the expression level of *csf1* by 3.63 times (Fig. 3j; Supplementary Fig. 10), which is nearly identical to the difference between *O. woworae* males and females (3.52 times higher in

males; Fig. 3d). The females administered with MT also had a significantly larger red-pigmented area in the pectoral fin than the control females (Supplementary Fig. 11). On the other hand, MT administration in females of *O. asinua*, a species that does not show red coloration in either sex, did not increase the *csf1*

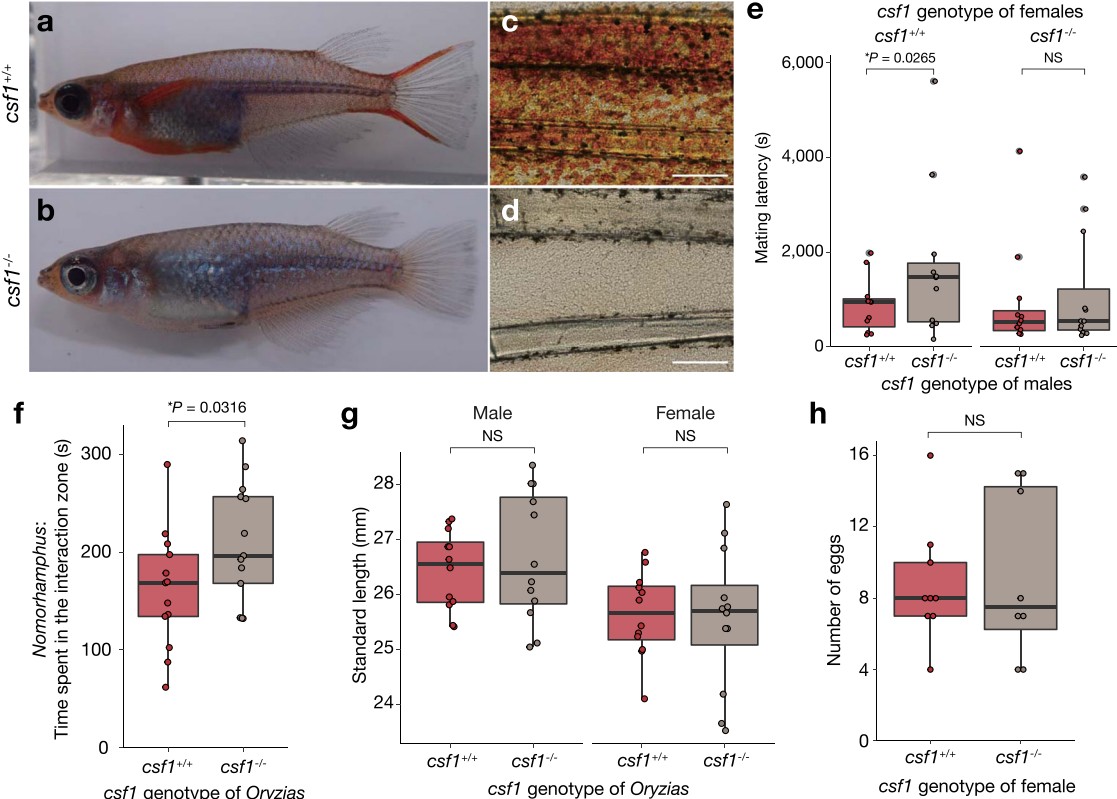

**Fig. 4 Role of *csf1* in red coloration and fitness components. a, b** Representative images of wild-type (*csf1*⁺/⁺) (**a**) and knockout (*csf1*⁻/⁻) *O. woworae* males (**b**). **c, d** Microscopic images of the pectoral fins of *csf1*⁺/⁺ (**c**) and *csf1*⁻/⁻ fish (**d**). Scale bars indicate 200 μm. Images are representative of three experiments. **e** Comparison of mating latency between *O. woworae* with *csf1*⁺/⁺ and *csf1*⁻/⁻. The mating behaviors were observed in 46 pairs: 11 pairs of *csf1*⁺/⁺ female and *csf1*⁺/⁺ male; 11 pairs of *csf1*⁺/⁺ female and *csf1*⁻/⁻ male; 12 pairs of *csf1*⁻/⁻ female and *csf1*⁺/⁺ male; 12 pairs of *csf1*⁻/⁻ female and *csf1*⁻/⁻ male. *$P < 0.05$ and not significant (NS) according to two-sided post hoc analysis with Tukey's multiplicity adjustment method in a generalized linear mixed model (GLMM). **f** Comparison of the behavioral preferences of male halfbeaks (*Nomorhamphus* cf. *ebrardtii*) for *csf1*⁺/⁺ and *csf1*⁻/⁻ *O. woworae* males (n = 13 halfbeaks). The X axis indicates the *csf1* genotype of *O. woworae* males and the Y axis indicates total time spent in each interaction zone by the halfbeaks during a 20-min behavioral test. The P-value (*$P < 0.05$) was calculated using the Wald test in a linear mixed model. **e, f** Each dot represents a single behavioral experiment. **g** No significant difference was found in the standard length between the *csf1*⁺/⁺ and *csf1*⁻/⁻ *O. woworae* adults that were used for the mating behavioral assay (n = 12 for each). Each dot indicates each individual. NS, not significant by two-way ANOVA. **h** No significant difference in the clutch size was found between the *csf1*⁺/⁺ and *csf1*⁻/⁻ *O. woworae* females (n = 9 for *csf1*⁺/⁺ and n = 8 for *csf1*⁻/⁻). **g, h** Each dot represents a single individual. NS, not significant according to the two-sided Mann–Whitney U test. **e–h** The red and gray colors indicate *csf1* wild-type and knockout fish, respectively. In the box-plots, the center line indicates the median, box limits indicate the upper and lower quartiles, the whiskers indicate 1.5× interquartile range, and the points are outliers. Source data are provided as a Source Data file.

expression in the pectoral fins (Supplementary Fig. 12). In addition, we found that fish with larger body sizes have higher expression levels of *csf1* in *O. woworae* (Supplementary Fig. 13), suggesting that *csf1* expression may change during growth. These data support that androgen signaling mediates the male-biased expression of *csf1* underlying red coloration in the pectoral fins of *O. woworae*. Comparison of putative androgen response elements (AREs) near *csf1* between *O. woworae* and *O. celebensis* revealed several variations in AREs in the non-coding regions (Supplementary Fig. 14).

**Red coloration in males affects female mate preference and predator preference.** Male-specific coloration is generally considered to be a courtship signal and to have an important role in female mate choice. We, therefore, investigated the effects of *csf1*-mediated male red coloration on female mate preference. Males and females of the wild-type (*csf1*⁺/⁺) and KO (*csf1*^Δ11/Δ11) genotypes were subjected to analysis of mating behavior. We tested all four types of pairs; i.e., between a wild-type female and a wild-type male (11 pairs), a KO female and a KO male (12 pairs), a wild-type female and a KO male (11 pairs), and a KO female and a wild-type

male (12 pairs). We first measured the time required for spawning (mating latency), which has been frequently used as an index of female receptivity in fishes, including the medaka fish[37]. A generalized linear mixed model (GLMM) showed that *csf1*-KO males spent significantly more time for spawning than wild-type males (GLMM, estimate ± s.e. = 0.56525 ± 0.20208, t = 2.797, $P = 0.00516$), regardless of whether the female was wild-type or *csf1*-knockout. Although we found a trend that the difference between the wild-type or *csf1*-knockout males was less clear when the female was *csf1*-knockout (Fig. 4e), neither the effect of female genotype (GLMM, estimate ± s.e. = −0.03979 ± 0.29268, t = 0.136, $P = 0.89187$) nor the interaction between the male and female genotypes (GLMM, estimate ± s.e. = −0.54946 ± 0.28436, t = −1.932, $P = 0.05333$) was significant. The KO mutation did not significantly affect the frequency of male approaches or the number of rejections by females (Supplementary Fig. 15). These results suggest that *csf1* genotype in males can influence female mate preference, which can contribute to reproductive success in males.

Male nuptial coloration often attracts not only conspecific females but also predators. We examined whether a halfbeak (*Nomorhamphus* cf. *ebrardtii*), a predator of *O. woworae*[38], is

attracted more to the wild-type red *O. woworae* males than non-red males. A male halfbeak was introduced into a test tank with two plastic tanks, one of which contained a wild-type male and another of which contained a KO male of *O. woworae* (Supplementary Fig. 16a). We measured the time spent by the halfbeak in front of each *O. woworae* (Supplementary Fig. 16b). A linear mixed model (LMM) with the genotype as a fixed factor demonstrated that the halfbeaks spent significantly longer time in front of the KO fish than the wild-type fish (Fig. 4f; LMM, estimate ± s.e.= 48.82 ± 20.07, $t_{12} = 2.432$, $P = 0.0316$). In addition, the halfbeaks significantly increased the number of entries into the interaction zone of the KO fish (Supplementary Fig. 16c), although the genotype did not show any changes in the latency to the first entry into each interaction zone (Supplementary Fig. 16d). These results suggest that, contrary to our expectation, the *csf1*-KO males with no red nuptial coloration are more attractive to predators than the wild-type red males.

In addition to the absence of a difference in survival among genotypes (see above), we found no significant differences in standard length (two-way ANOVA, genotype: $F_{1,45} = 0.082$, $P = 0.7765$, sex: $F_{1,45} = 10.883$, $P = 0.0022$; Fig. 4g) or female fecundity (i.e., egg production) (Mann–Whitney $U$ test, $W = 38.5$, $P = 0.8222$; Fig. 4h) among the genotypes.

## Discussion

In this study, we have established a reference genome sequence and a platform for targeted genome editing in Sulawesian medaka fishes. Given that they show great phenotypic diversity (Fig. 1), that some of the species are interfertile and can be crossed for linkage mapping, and that the genome size is relatively small (~690 Mb), the Sulawesian medaka fishes will be a great model to explore the evolutionary genetic mechanisms underlying phenotypic diversification, including sexual dichromatism[29]. Furthermore, several Sulawesian medaka species are symparic and have undergone interspecies hybridization[39], making them a good model system for studying the mechanisms of speciation and hybridization, although *O. celebensis* and *O. woworae* used in this study are allopatric with no hybrids observed in the wild.

In the present study, integration of genomic analysis and genome-editing technology has revealed that a *cis*-regulatory mutation at an autosomal gene *csf1* caused the acquisition of sexual dichromatism in pectoral fins of *O. woworae*. Gene expression of *csf1* was under the control of androgen, which enables male-specific expression of red coloration. These results provide compelling evidence that evolutionary changes in androgen signaling pathways can cause the evolution of sexual dimorphism[20,22,23]. Previously, we have proposed that androgen-dependent regulation contributes more to age-specific expression of sexual dimorphism than sex-linkage, whereas sex-linkage is more important for sexual dimorphism expressed throughout life[25]. Given that sexual dichromatism appears only at adult stages in *O. woworae*, our present result supports this hypothesis.

The application of genome-editing enabled us to investigate the contributions of *csf1* to several fitness components. Gene knockout did not affect intrinsic viability, body size, or female fecundity. Knockout males, however, required more time for spawning with females (Fig. 4e), indicating that they are less attractive to females than wild-type males. This suggests that *csf1*-mediated red coloration may help to increase male reproductive success, in accordance with sexual selection theories generally positing that male ornaments increase reproductive success[1,2]. Sexual selection theories also often posit that male ornaments increase the possibility of detection by predators and increase predation risks[11,12]. Contrary to this hypothesis, we found that *csf1*-knockout non-red males attracted predatory

halfbeaks more than control red males (Fig. 4f). Previous studies in some predator–prey systems have also shown that individuals with ornamental colorations were attacked by the predators less frequently than individuals with less ornaments[40–42]. This may be because male ornaments are often indicators of good physical conditions[2] and higher escape abilities[43]; therefore, predators may select less-colorful males with higher vulnerability to predation rather than colorful males with higher escape abilities[44,45]. These results indicate that functional analysis of causative genes underlying sexually selected traits by gene manipulation provides us a new avenue for experimentally testing hypotheses upon which sexual selection theories are based.

Identification of causative genes also enables us to understand whether the same genes are repeatedly used for convergent evolution. Previous studies on adaptive traits have shown that the same genes or genes in the same pathways are repeatedly used for convergent evolution[46–48]. In contrast, we know little about the genetic basis for the convergent evolution of sexually selected traits. Red male ornaments have evolved in many different lineages of fishes, including sticklebacks[49], cichlids[50], and bluefin killifish[51]. Currently, we do not know whether *csf1* is also involved in sexually dimorphic red coloration in other fishes. However, it is possible that *csf1* or the *csf1*-signaling pathway is involved in sexual dichromatism in other fishes. In the zebrafish (*Danio rerio*), csf1 signaling is known to be essential for the development of xanthophores[34], and transgenic misexpression of *csf1* promoted regionally specific development of xanthophores[35]. This is consistent with our observations that differential expression patterns of *csf1* match differential patterns of red/yellow coloration. In fact, upregulation of *csf1* in skin by a *cis*-regulatory mutation increases xanthophores that causes the uniform pattern of skin pigmentation in a related species of the zebrafish (*D. albolineatus*)[52]. In addition, a QTL controlling unusual red throat coloration in female threespine stickleback (*Gasterosteus aculeatus*) contained *csf1*[53]. Further genetic analysis of sexual dichromatism in other taxa will help to understand whether there are any hotspot genes or hotspot pathways that respond to sexual selection.

Several environmental factors of the habitats may contribute to the evolution of the red coloration in *O. woworae*. The natural habitat of the *O. woworae* population used in this study is dominated by light with shorter-wavelengths[38]. Such environments generally create blue background and make males with red colorations more conspicuous to conspecific females[51]. Such sensory drive may promote the evolution of red coloration in *O. woworae*. Behavioral analysis of female mate preference and predation experiments under manipulated light conditions, which have been successfully conducted in other fishes[54,55], will lead to a better understanding of the environment-dependent fitness effects of the *csf1* gene in *O. woworae*. Analysis of female preference for red coloration in other species that do not show red coloration is also necessary.

Although female mate preference is one of the key factors for the evolution of male mating signals[1,2,56], we currently know little about the molecular genetic basis for female mate preference. Our data showed a trend that *csf1*-knockout females have less preference for red males compared to the wild-type females (Fig. 4e). This suggests that *csf1* may control not only male mating signals but also female mate preference. Theoretical studies indicate that the linkage of male mating signals and female mate preference can promote the evolution of exaggerated male ornamental traits[20,57,58]. As CSF1 deficiency in neuronal cells is shown to alter neuronal functions and the social preference in mice[59], it is possible that *csf1* has a pleiotropic effect on both pigment cell and brain development.

There are several caveats to the present study. First, we have not measured total fitness in a natural setting. *csf1* is known to

regulate differentiation and regulation of several types of cells, such as macrophage[60–63], osteoclasts[64,65], and microglia[63,66,67]. Therefore, it is possible that *csf1* may also influence immunity, skeletal morphology, and behavior, although we did not observe increased lethality or apparent deformity in the knockout fish. Measurement of total fitness and changes in allele frequencies over generations in a more natural setting would be possible in the future using mesocosms. Second, we have not yet identified the causative mutations in the *cis*-regulatory region. Interestingly, although *csf1* expression in the pectoral fin of *O. woworae* males was unique in the Sulawesian medakas examined thus far, *csf1* was expressed in pelvic and caudal fins in males of both *O. woworae* and its related species *O. asinua* (Fig. 3e). Currently, we do not know what kinds of mutations enabled the androgen-dependent expression of *csf1* only in the pectoral fin. Identification of causative mutations would be possible using gene replacement technologies of CRISPR/Cas, which has been already successful in the Japanese medaka fish[68].

In conclusion, we have demonstrated that integration of genomic analysis with genome-editing technologies can promote the identification of causative genes underlying divergence in sexually selected traits. Furthermore, genetic manipulation enables us to test several hypotheses of sexual selection theories. Further integrative genomic studies on sexually selected traits in multiple taxa will improve our understanding of the evolution of biodiversity driven by sexual selection.

## Methods

**Ethics statement for animal experiments.** All animal experiments were conducted under approval by the Institutional Animal Care and Use Committee of the National Institute of Genetics (28-12, 27-5, and 26-15) and the National Institute for Basic Biology (19A055, 18A079, and 17A161).

**Reference genome sequencing and assembly.** To create a reference assembly for Sulawesian Adrianichthyidae, we sequenced the genome of *O. celebensis*; a closed colony of this species (Ujung pandang strain) that had been maintained in the laboratory for ~40 years and is expected to have low heterozygosity. This laboratory stock of *O. celebensis* strain was originally collected in Makassar, Sulawesi, Indonesia by Drs. K. Hirata and T. Iwamatsu on March 1979[69] and was provided for this study by the National Bioresource Project (NBRP) medaka (RS278; https://shigen.nig.ac.jp/medaka/).

High-molecular-weight genomic DNA was extracted from the skeletal muscle of a female with a QIAGEN Genomic-tip (Qiagen, Hilden, Germany). First, 250-bp paired-end sequencing was conducted using Illumina HiSeq 2500 with a TruSeq DNA PCR-free Sample Prep Kit (Illumina, San Diego, CA), and high-coverage short-read sequences were obtained (`296 million reads corresponding to 148 Gb with ~185× coverage). K-mer analysis estimated a heterozygosity of 0.2–0.3%. Next, a library with inserts >20 kb was generated using the SMRTbell Template Prep Kit 1.0 (Pacific Biosciences, Menlo Park, CA) and then sequenced by PacBio RSII (Pacific Biosciences) using 104 single-molecule real-time (SMRT) cells with P6/C4v2 chemistry. A total of 7.5 million reads (88.1 Gb; ~110× coverage) were obtained and assembled into contigs by the assemblers Falcon v0.7 and Falcon Unzip v0.4.0 (https://github.com/PacificBiosciences/FALCON). To correct the sequences, the Illumina short reads were mapped to the contigs using BWA-MEM v0.7.7[70] and polished using samtools v0.1.9[71]. The polished contigs were scaffolded by optical mapping using the Iris system with Bionano Solve v3.1 (two-hybrid scaffolding: BspQI-BbvCI) (Bionano Genomics, San Diego, CA). The final assembly included 688 Mb consisting of 631 scaffolds (2633 contigs) with high continuity (scaffold N50 length: 21.5 Mb and contig N50 length: 6.1 Mb).

The completeness of the assembly was evaluated using Benchmarking Universal Single-Copy Orthologs (BUSCO) v3.1.0[72] based on 4584 BUSCOs in the lineage data set actinopterygii_odb9, which exhibited high completeness: 4443 (96.9%) of complete BUSCOs, 48 (1.0%) of fragmented BUSCOs, and 93 (2.1%) of missing BUSCOs.

**Chromosome-level assembly of the *O. celebensis* genome.** To anchor the scaffolds to chromosomes, a linkage map was created using an $F_2$ fish between *O. woworae* and *O. celebensis*. An *O. woworae* male caught at Fotuno Fountain was first crossed with an *O. celebensis* female caught at Malino River to produce $F_1$ fish. Next, an $F_1$ female was crossed with an $F_1$ male, and 164 $F_2$ adult fish (91 males and 73 females) were obtained. To create a linkage map, all $F_2$ fish were genotyped using double-digest restriction site associated DNA (ddRAD). Genomic DNA was extracted from the muscle tissue of the adult fish that were stored in RNAlater

(Thermo Fisher Scientific) using the DNeasy Blood & Tissue Kit (Qiagen). The libraries were prepared as described previously[73–75]. In brief, 10 ng of genomic DNA was digested with *Eco*RI and *Bgl*II, and adapters were ligated into the digested sites. The ligated fragments were amplified by PCR using an index primer and the TruSeq universal primer, followed by the purification of fragments of ~320 bp. The libraries were sequenced with the Illumina HiSeq2000 (single-end 50 bp) at Macrogen Japan (Kyoto, Japan). In total 17,489,472 and 10,699,988 reads (891,963,072 and 545,699,388 bp) were obtained from the *O. woworae* $G_0$ male and the *O. celebensis* $G_0$ female, respectively. For $F_2$ progenies, 1,893,237 ± 781,375 reads were obtained on average (±SD).

The reads were trimmed by Trim Galore 0.4.3 (https://www.bioinformatics.babraham.ac.uk/projects/trim_galore/) with Cutadapt 1.12 (https://cutadapt.readthedocs.io/) and mapped to the *O. celebensis* reference by STAMPY v1.0.32[76] with a 3% substitution rate allowed. The mapped reads were converted to sorted bam files using *view* and *sort* in samtools v1.7[71], and variant bases were only called from uniquely mapped reads (MQ > 10) using *mpileup* and *call* in bcftools v1.7[77]. Then, indels and SNPs were removed if they had a low genotyping quality (GQ < 20), a low read depth (DP < 5), a low frequency of the minor allele (<5%), or more than four alleles or were missing in >10% of individuals in the family using vcftools v0.1.15[78]. The vcf file was converted to the raw file using the *vcf2raw* function in the R package onemap (https://github.com/augusto-garcia/onemap), and 1865 informative markers were finally kept for the linkage analysis.

A linkage map was constructed using this genotype data with the R package qtl[79]. Duplicated markers and markers showing significant segregation distortion ($\chi^2$ test, $P < 0.05$, Bonferroni correction) were excluded. LGs were formed with a minimum LOD score of 6 and a maximum recombination fraction of 0.35. The order of the markers in each LG was estimated using the *orderMarkers* function. After the initial ordering, the recombination fractions and LOD scores of all pairs in each LG were plotted, and suspicious orders were corrected using the *orderMarkers* function or manual correction. Changes in the length of each LG by dropping one marker were estimated with the *droponemarker* function, and then the markers that decreased the length by more than 5 cM were removed. Possible genotyping errors were identified by calculating LOD scores with the *calc.errorlod* function, and genotypes with error LOD scores above three were excluded. Again, markers showing significant segregation distortion ($\chi^2$ test, $P < 0.001$) were removed. Finally, a linkage map was constructed consisting of 18 LGs at 1484.6 cM with 1281 markers.

Chromosomal sequences were reconstructed from the assembled scaffolds and the linkage map using ALLMAPS[80]. In total, 57 scaffolds (635.18 Mb; 92.3%) were assigned into 18 LGs by 1,278 markers and 35 scaffolds (577.51 Mb; 83.9%) were oriented by 1169 markers.

**Repeat masking.** The de novo repeat-finding package RepeatModeler v1.0.11 (http://www.repeatmasker.org/RepeatModeler/) was used to identify and classify repetitive sequences in the *O. celebensis* reference genome. A total of 1472 repeat consensus sequences were found, of which 779 (52.9%) were classified into the known repeat families. The repetitive regions of the *O. celebensis* reference assembly were masked by RepeatMasker v4.0.7 with the de novo repeat library and RMBlast v2.6.0. Low-complexity DNA and simple repeats were left unmasked with the "-nolow" option. After this process, 204.0 Mb (30.22%) of the genome was masked.

**Transcriptome sequencing and gene annotation.** For gene annotation, RNA-seq was conducted on seven tissues (brain, eye, gill, liver, gonad, kidney, and spleen) of one adult male and one adult female and whole embryos at two developmental stages (3 days and 6 days after fertilization) in laboratory-raised *O. celebensis*. Total RNA was isolated using the RNeasy Universal Plus Mini Kit or an RNeasy Micro Kit (Qiagen), and the concentrations of the isolated RNAs were measured using a Quant-iT RiboGreen RNA Assay Kit (Thermo Fisher Scientific, Waltham, MA). After mRNA isolation with the NEBNext Poly(A) mRNA Magnetic Isolation Module (New England Biolabs, Ipswich, MA), sequencing libraries with 250–400 bp inserts were prepared using the NEBNext Ultra RNA Library Prep Kit for Illumina and NEBNext Multiplex Oligos for Illumina (Dual Index Primers) (New England Biolabs). Sequencing was performed using the Illumina HiSeq 4000 (2 × 100 bp) at Macrogen Japan (Kyoto, Japan).

The obtained reads were trimmed using Trim Galore 0.4.3 and Cutadapt 1.12, and the trimmed reads were mapped to the repeat-masked reference assembly by HISAT2 v2.1.0[81]. The mapped reads were converted to bam files and sorted using samtools. StringTie v1.3.5 was used to assemble potential transcripts for each sample and generate gtf files[82]. Then, all assemblies were merged using TACO v0.7.3[83] to generate a merged gtf file, which contained 56,305 transcripts in 27,468 loci. For each transcript, the open reading frame (ORF) was predicted using TransDecoder v5.5.0 (https://transdecoder.github.io/), and peptide sequences with 30 or more amino acids were subjected to the following analysis. A single ORF was retained per transcript that contained the longest ORF and homology with a protein sequence of *O. latipes* (assembly: ASM223467v1), *O. malastigma* (assembly: Om_v0.7.RACA) or *Danio rerio* (assembly: GRCz11) in Ensembl Release 94 (http://www.ensembl.org/) based on BLASTP searching (e value < 1e−10) or pfam-A database search with *hmmscan* in HMMER v3.2.1 (http://hmmer.org/).

The final gene models contained 24,120 coding genes with 51,523 transcripts. Transcriptome completeness was estimated again using BUSCO v3.1.0 based on 4584 BUSCOs in the lineage dataset actinopterygii_odb9: 4304 (93.9%) of complete BUSCOs, 144 (3.1%) of fragmented BUSCOs, and 136 (3.0%) of missing BUSCOs.

**Gene synteny between O. celebensis and O. latipes**. To visualize the conserved synteny between the Sulawesian medaka and the Japanese medaka, the peptide sequences of the O. celebensis gene data set (see above) and the Japanese medaka (O. latipes Hd-rR; ASM223467v1) from Ensembl Release 94 (http://www.ensembl.org/) were analyzed. A reciprocal BLAST analysis was performed with OrthoFinder v2.2.6[84], and 13,777 single orthologs were identified. The positions of each ortholog on the genome of the two species were connected as a line using Circos v.0.69-6[85].

**Phylogenomics of Sulawesi-endemic species**. For the phylogenetic analysis of Sulawesian medakas, whole-genome sequencing was conducted for 26 wild-caught fish of 17 species from 20 sampling sites (Fig. 1a; Supplementary Table 1). Genomic DNA was extracted from the muscle tissue of the adult fish using the DNeasy Blood & Tissue Kit (Qiagen). For one O. woworae caught at Fotuno Fountain, the library was constructed using the Illumina TruSeq DNA Sample Prep Kit (Illumina) and sequenced with the Illumina HiSeq2000 (2 × 100 bp) at Takara Dragon Genomics (Mie, Japan). For other fishes, sequencing libraries were prepared using the NEBNext Ultra DNA Library Prep Kit for Illumina with NEBNext Multiplex Oligos for Illumina (New England Biolabs) and sequenced with the Illumina HiSeq X-Ten (2 × 150 bp) at Macrogen Japan.

The sequenced reads were trimmed using Trim Galore 0.4.4_dev with Cutadapt 1.17 and mapped to the O. celebensis reference by BWA-MEM v0.7.17[70]. The bam files were exported and sorted using SortSam in Picard Tools (https://broadinstitute.github.io/picard/). The PCR duplicates in each alignment were removed using MarkDuplicates in Picard Tools. Uniquely mapped reads (MQ > 10) were piled up by bcftools v1.9 mpileup, and then bases at both variant and invariant sites were called across the O. celebensis reference genome with bcftools call. Bases were filtered out with an insertion or deletion, a Phred quality score <20, a very low read depth (DP < 5), or very high coverage (DP > 200) using bcftools view.

Reference genome assemblies and gene annotations of O. latipes (ASM223467v1)[86], O. sakaizumii HNI (ASM223471v1)[86], O. latipes HSOK (ASM223469v1)[86], O. sinensis (ASM858656v1), O. malastigma (Om_v0.7.RACA)[87], O. javanicus (OJAV_1.1)[88], and Xiphophorus maculatus (X_maculatus-5.0-male) were obtained from Ensembl Release 100 (http://www.ensembl.org/). The longest transcript was selected as a representative for each gene, and a coding and a peptide sequence were extracted from each. Orthologous groups among the O. celebensis and seven other assemblies were classified using OrthoFinder v2.3.11[89]. A total of 10,188 single-copy orthologous genes were identified from the OrthoFinder results and used for the subsequent phylogenetic analysis.

The coding region of each single-copy gene was extracted from the merged vcf file of Sulawesian fishes using the Perl script vcf-to-tab in VCFtools v0.1.16[78] and a custom script (https://github.com/satoshi-ansai/genome_utils/blob/master/vcf-tab_to_fasta.py) written with Python 3.7.6 as well as from the eight reference assemblies. After removing the orthologs with no known nucleotides in at least one of the analyzed taxa, codon alignment based on translated peptide sequences for each gene was generated by MACSE v2.03[90]. The alignment sequences of 10,174 genes with a total of 18,813,756 sites were concatenated and converted into a single phylip file using AMAS concat[91] by setting each single copy as a separate partition. A maximum-likelihood tree was estimated using IQ-TREE v1.6.12[92] with the codon-specific GTR+G models, and the reliability of each node was assessed by an ultrafast bootstrap analysis of 1000 replicates[93]. The phylogenetic tree rooted in X. maculatus was drawn using FigTree v1.4.4 (https://github.com/rambaut/figtree/) and then edited by Adobe Illustrator 2020 (Adobe, San Jose, CA).

**Divergence time estimation**. For the divergence time estimation using the whole-genome sequence data, "clock-like genes" were extracted from 10,174 genes of the phylogenomic data sets using SortaDate[94]. First, a maximum-likelihood gene tree was estimated for each ortholog by RAxML v8.2.12[95] using the codon-specific GTR+G models with a rapid bootstrap analysis of 100 bootstrap replicates. Next, the top 500 clock-like genes were filtered using scripts in SortaDate with the following three criteria: (1) low root-to-tip variance as an indication of clock-likeness, (2) small conflict with a concatenated tree estimated using 10,174 genes (Fig. 1b), and (3) reasonable tree length for obtaining discernible information. A time-calibrated tree was inferred using the merged sequence of the 500 clock-like genes (total 1,830,357 sites) by the RelTime method[96,97] using the maximum-likelihood method with the GTR+G model in MEGA X[98]. As a calibration point, the opening of the Makassar Strait, ca. 45 million years ago (Mya)[99–101] was employed for the node at which the Sulawesi fishes split from the O. javanicus group; practically, a normal distribution with $\mu = 45$ Mya and $\sigma = 3.0$ (95% CI = 39.12–50.88) was set as the calibration time. The time-calibrated tree was drawn using FigTree v1.4.4 and edited with Adobe Illustrator 2020 (Adobe).

**QTL mapping**. The $F_2$ family (91 males and 73 females) derived from an O. celebensis female and an O. woworae male that was used to construct the linkage map (see above) was also used for the QTL analysis. For phenotyping, each $F_2$ fish

was anesthetized with ethyl 3-aminobenzoate methanesulfonate (MS-222, Sigma-Aldrich). After the left side of each fish was photographed with a digital camera (Optio W90, Pentax, Tokyo), the reflectance of the pectoral fin and lateral body was measured using a spectrophotometer (USB2000, Ocean Optics, Dunedin, USA) in triplicate. The reflectance was analyzed with the R package pavo[102] as follows: first, the reflectance values with wavelength ranging from 300 to 700 nm were averaged over the triplicates and smoothed by Loess regression. Further, the relative contributions of a particular spectral ranges to the total brightness (S1) were calculated as chroma associated with specific hues: S1.Violet: 300 –415 nm; S1.Blue: 400–510 nm; S1.Green: 510–605 nm; S1.Yellow: 550–625 nm; S1.Red: 605–700 nm. For the binary analysis of coloration traits, $F_2$ fish were classified according to the presence or absence of pigment cells (red pigment cells in the dorsal side of the pectoral fin and iridophores in the lateral body) and black stripes of melanophores in the tail fin. The standard length of each fish was measured using ImageJ[103].

For QTL analysis, the R package qtl[79] was used. First, genotype probabilities were estimated using the calc.genoprob function, and imputation was performed using the sim.geno function. For binary and quantitative traits, interval mapping using the scanone function was performed using the EM algorithm with the binary model and the multiple imputation method, respectively. Sex was included as an additive covariate in the QTL analysis of coloration traits. The significant thresholds of the logarithm of the odds (LOD) scores were determined by 1000 genome-wide permutations (significant threshold: $\alpha = 0.05$). The 95% Bayesian CI of each significant QTL was calculated using the bayesint function. The percentage of phenotypic variance explained was estimated, and the additive ($a$) and the dominance ($d$) effects of each significant QTL were determined using the fitqtl function.

**Transcriptome analysis of pectoral fins**. O. woworae caught at Fotuno Fountain and O. asinua caught at Asinua River (four males and four females per species) were used for RNA-seq analysis. Total RNA was extracted from the pectoral fin using an RNeasy Micro Kit (Qiagen). RNA quantification and sequencing library preparation were performed with the same procedure as described in the section "Transcriptome sequencing and gene annotation". The libraries were sequenced using the Illumina HiSeq 4000 (2 × 150 bp) at Riken genesis (Yokohama, Japan).

The obtained reads were trimmed using Trim Galore 0.4.3 with Cutadapt 1.12 and mapped on the O. celebensis reference assembly using STAR v2.6.1d with default parameters. The mapped read numbers of each gene were counted using featureCounts v1.6.3[104]. The following statistical analysis was performed using the R package edgeR v3.22.5[105]. The raw read counts for genes with at least one count per million in at least four samples of each analysis (16,072 genes for analysis of differentially expressed genes between sexes and 16,304 genes for analysis of differentially expressed genes between species) were used for the subsequent analysis. After normalization with the trimmed mean of M-values method, differentially expressed genes were identified by Fisher's exact test with FDR of 1%.

**Phylogenetic analysis of csf1 gene**. Coding sequences of csf1 across 18 Actinopterygian fishes and non-Actinopterygian vertebrates (Homo sapiens and Gallus gallus) were collected using ORTHOSCOPE v1.0.2[106]. The csf1 gene sequence of O. woworae was used as a query. The csf1 coding sequences of five Oryzias species were also identified in Ensembl Release 100 (O. latipes Hd-rR: ENSORLT000000 13159.2, O. sakaizumii HNI: ENSORLT00020035416.1, Oryzias sp. HSOK: ENSO RLT00015023569.1, O. sinensis: ENSOSIT00000040766.1, and O. javanicus: ENSO JAT00000045443.1) and of O. celebensis in the in-house gene annotation (TU49 810.p1). The translated protein sequences were aligned using MAFFT v7.453[107], and the corresponding coding nucleotide sequences were aligned using PAL2NAL v14[108]. The aligned sequences were trimmed by removing poorly aligned regions using trimAl v1.4.rev15[93] with the "gappyout" option. The maximum-likelihood tree was inferred using a total of 684 bp in the alignment with iqtree v1.6.12 using the codon-specific GTR+G model[92]. The reliability of each node was assessed by an ultrafast bootstrap analysis of 1000 replicates[109].

**Identification of androgen response elements**. A genomic sequence between the predicted transcription start site of a neighboring gene at the upstream of csf1 (ren) and that of another neighboring gene at the downstream (fmodb) was extracted from each genome assembly of O. woworae (34,005 bp) and O. celebensis (33,555 bp). To this end, we conducted de novo assembly of O. woworae genome. Genomic DNA was extracted from the whole body of a laboratory-raised strain of O. woworae (Fotuno strain). In total 18.95 Gb of genomic reads (1.36 M reads, average read length 13,883 bp) were obtained by SMRT sequencing with PacBio Sequel system (Pacific Biosciences). We assembled these reads using Canu v1.7[110] with the parameter correctedErrorRate = 0.105, an optimized value for low coverage (20×) data sets. Then, the raw SMRT reads were mapped to the assembled contig with pbalign v0.3.1 (https://github.com/PacificBiosciences/pbalign) and the contig sequences were polished with arrow v2.2.2 (https://github.com/PacificBiosciences/GenomicConsensus). We also mapped the Illumina short reads of a wild-caught female from Fotuno Fountain (Supplementary Table 1) to the polished contigs with BWA-MEM v0.7.17 and improved the sequence accuracy using Pilon v1.22[111]. We used an androgen receptor-binding motif (GNACANNNWG) predicted from human ChIP-seq data in JASPAR[112] as a possible androgen response element

(ARE) in the medaka. The positions of ARE on each sequence were identified using 'findMotifGenome.pl' in HOMER v4.11[113]. The two genome assemblies were aligned based on BLAST hits (BLASTN, v2.6.0+), and then the positions of genes and AREs were plotted using Easyfig v2.2.5[114].

**Androgen administration.** A synthetic androgen MT (132-09933, Fujifilm Wako Chemicals, Osaka, Japan) was administered to adult *O. woworae* females (Fotuno strain) or *O. asinua* females for 14 days following a previous method[115]. In brief, the female medakas were exposed to 32 nM MT or solvent (0.01% dimethyl sulfoxide, DMSO) as a control at a density of six fish per 2 L of tank water. Fresh solutions of MT or DMSO were added every 2 days.

Each fish was anesthetized using MS-222 and then the left side was photographed using a digital camera Tough TG-6 (Olympus, Tokyo, Japan). For each left pectoral fin, the total area of the fin and the area with red pigmentation were measured using ImageJ[103].

**Quantitative RT-PCR.** Total RNA was extracted from the fin using an RNeasy Micro Kit (Qiagen) and quantified using the Quant-iT RiboGreen dsDNA Assay Kit (Thermo Fisher Scientific). Following DNA elimination by DNaseI (Thermo Fisher Scientific), the first-strand cDNA was synthesized with a High Capacity cDNA Reverse Transcription Kit (Thermo Fisher Scientific) using 200 ng of total RNA as a template for pectoral fins from the $F_2$ family and 100 ng of total RNA for all others. The *rpl13a* gene was also analyzed as an internal control, which showed stable expression under different conditions and was used as an internal control for the qPCR experiment in stickleback[116]. Primer pairs for *csf1* and *rpl13a* were designed with Primer Express 3.0 software (Thermo Fisher Scientific) (Supplementary Table 5). A quantitative PCR analysis was performed using Fast SYBR Green Master Mix (Thermo Fisher Scientific) on a StepOnePlus Real-time PCR system (Thermo Fisher Scientific). Relative expression levels were calculated from standard curves drawn from serially diluted cDNA pools of all analyzed fish in each plate.

For analysis of the effects of MT administration (Supplementary Fig. 12) and body size (Supplementary Fig. 13) on the expression levels of *csf1* in *O. asinua* females and *O. woworae* fish (both sexes), respectively, the first-strand cDNA was synthesized with ReverTra Ace qPCR RT Master Mix with gDNA Remover (Toyobo, Osaka, Japan) using 50 ng of total RNA as a template. A quantitative PCR analysis was performed using THUNDERBIRD Next SYBR qPCR Mix (Toyobo) on a CFX Connect Real-time PCR detection system (Bio-Rad, Hercules, USA) with the same primer pairs as described above.

**Allele-specific expression analysis.** Allele-specific expression analysis of *csf1* was conducted as described previously[117]. A region of the fourth exon of the *csf1* gene containing the species-specific SNP was amplified from the pectoral fin cDNA of *O. celebensis* × *O. woworae* $F_2$ fish that were heterozygous at the *csf1* locus. As a control, genomic DNA was also analyzed. PCR reactions were conducted using KOD -Plus- Neo (Toyobo) and the primers csf1-exon2-FW and csf1-qPCR-RV (Supplementary Table 5) with the following conditions: 1 cycle of 94 °C for 2 min and 35 cycles of 98 °C for 10 s, 60 °C for 20 s, and 68 °C for 30 s followed by maintenance at 4 °C. The products were treated with Illustra ExoProStar to eliminate primers (GE Healthcare Life Sciences, Chicago, IL) and sequenced by a Sanger sequencer at Eurofins Genomics (Tokyo, Japan). Sequence chromatograms were visualized using ApE v2.0.53 (http://jorgensen.biology.utah.edu/wayned/ape/), and the regions containing the species-specific SNP were stored as EPS files. Each peak ("C" for the *O. woworae* allele and "A" for the *O. celebensis* allele) at the SNP site was extracted and converted into JPEG image files using Adobe Illustrator CS6 (Adobe). The number of pixels in each peak was counted using Adobe Photoshop CS6 (Adobe), and the C/A ratio was calculated both in gDNA and cDNA data. Differences were analyzed in the C/A ratio between the gDNA and cDNA templates using Wilcoxon's signed rank test in the R package *exactRankTests*.

**Genome editing.** To investigate the functions of *csf1* in vivo, CRISPR/Cas-induced knockout and knock-in fish were generated. RNAs for CRISPR/Cas were prepared according to a previously described method[118] with slight modifications. In brief, capped RNA for the Cas9 nuclease was synthesized using the pCS2+hSpCas9 vector as a template with the mMessage mMachine SP6 Kit (Thermo Fisher Scientific). To synthesize sgRNAs, 57-mer oligonucleotides containing a T7 promoter sequence and an 18-mer custom target sequence were designed (Supplementary Table 5). The sgRNA templates were PCR-amplified from pDR274 vectors[119] using the oligonucleotides and primer sgRNA-RV (Supplementary Table 5) with KOD -Plus- Neo (Toyobo) and were purified using the NucleoSpin Gel and PCR Clean-up Kit (MACHEREY-NAGEL, Düren, Germany). Next, sgRNAs were synthesized using the AmpliScribe T7-Flash Transcription Kit (Epicentre, Madison, WI). The transcribed RNAs were purified using the RNeasy Mini Kit (Qiagen) and their concentrations were measured using the spectrophotometer NanoDrop Lite (Thermo Fisher Scientific). For the GFP knock-in experiment, a donor DNA plasmid (pBS-Tbait-olhs-GFP) and an sgRNA for the Tbait sequence were prepared according to a previous method[120].

Microinjection into *O. woworae* fertilized eggs was performed following a previously established method for *O. latipes*[121] with slight modifications. In brief, a

new mold was made to create egg holders of 1.1-mm width for *O. woworae* eggs because the egg size of *O. woworae* is larger than that of *O. latipes*. In addition, fertilized eggs were maintained at room temperature instead of on ice because *O. woworae* eggs showed higher mortality on ice than at room temperature, although incubation on ice is frequently used for *O. latipes* eggs to slow down the developmental speed and allow more time for the injection procedure. For targeted knockout, 2–3 nL of mixtures containing 100 ng/µL of Cas9 RNA and 50 ng/µL of the sgRNA for *csf1* #2 were injected. For GFP knock-in, we injected ~1 nL of mixtures containing 100 ng/µL of Cas9 RNA, 10 ng/µL of the sgRNA for *csf1* #1, 10 ng/µL of the sgRNA for Tbait, and 5 ng/µL of the donor plasmid were injected.

Genotyping of the mutant fish was conducted by a heteroduplex mobility assay using an automated electrophoresis system (MultiNA MCE-202; Shimadzu, Kyoto, Japan)[122] with the primers csf1-exon1-HMA-FW and csf1-exon1-HMA-RV (Supplementary Table 5). The sequences of mutant alleles isolated from $F_1$ or later generations were determined by direct Sanger sequencing of the PCR products using the primer pair csf1-exon1-HMA-FW (Supplementary Table 5).

**Measurement of reflectance spectra in *csf1*-knockout fish.** After anesthesia using MS-222, the reflectance of the left pectoral fin was measured with a spectrophotometer (USB2000, Ocean Optics) in triplicate. The spectral data were analyzed using the R package *pavo*[102]. As an index of red pigmentation, the relative contribution of a red spectral range (605–700 nm of wavelength) to total brightness (S1.Red) was calculated in each sample. For principal component analysis of spectral shape, each spectral data were first processed to correct spectra to have a mean reflectance of zero and then binned into the width of 20 nm.

**Microscopic observation.** GFP fluorescence in the whole bodies of adult fish was observed using a stereo fluorescent microscope SZX16 (Olympus). The images were captured using a digital camera DP71 (Olympus). The detailed tissue structures of the fins were observed using an inverted microscope IX81 (Olympus) with a digital camera DP72 (Olympus). The brightness and contrast of each image were adjusted using Fiji of ImageJ[123].

**Mating behaviors.** Heterozygous females and males of a *csf1*-KO strain (*csf1*$^{+/Δ11}$) were crossed. Genomic DNA was extracted from a fin clip of each progeny before sexual maturation, and the genotype of each fish was determined by a heteroduplex mobility assay as described above. To exclude the possibility that females choose familiar males with particular genotypes, individuals with different genotypes were reared together in a single 3-L tank: i.e., a single rearing tank contained one wild-type (*csf1*$^{+/+}$) male, one wild-type (*csf1*$^{+/+}$) female, one homozygous mutant (*csf1*$^{Δ11/Δ11}$) male and one homozygous mutant (*csf1*$^{Δ11/Δ11}$) female. Therefore, the tested females were familiar with both wild-type and mutant males. They were subjected to analysis of mating behavior at 6 months post hatching. One day before each mating experiment, a male and a female were randomly selected from the two different tanks and transferred to a small acrylic tank (200 mm length × 75 mm width × 150 mm height) consisting of black panels except for a transparent acrylic panel at the front face and filled with 2 L water at 26 °C. The tank was illuminated by a white LED light (LT-N10S-D, Ohm Electronic, Tokyo, Japan), whose light spectrum (Supplementary Fig. 17) and intensity (20.44 µmol m$^{-2}$ s$^{-1}$; 1387 Lux) were obtained using a light analyzer (LA-105, NK System, Osaka, Japan). The female was separated in a transparent plastic cup with small holes to exchange water. The next morning (at 7:00–8:00 AM), the behavioral trial was initiated by removing the cup. Their behavior was recorded for 2 h using the Raspberry Pi 3 modelB with the NoIR Camera Module V2 (Raspberry Pi Foundation, Cambridge, UK). After the trial, the fish were returned to their original tanks. Each female was tested with a wild-type male and a mutant male on different days in random order: each fish was tested only twice in total. Fish that did not spawn eggs during the 2 h-behavioral test were excluded from the subsequent analysis.

The interval between the removal of the separator and successful spawning was measured as the mating latency. In successful spawning, a male holds a female with median fins, and both fish quiver; this is followed by female egg spawning within a few seconds. Therefore, the time until both fish started quivering was measured. The numbers of male approaches toward females and rejections by females after being held by the male were also counted from the recordings. Statistical analysis was conducted using R version 4.0.0 with GLMMs by the "glmer" function in the package *lme4* version 1.1-23. A gamma distribution and a Poisson distribution with a log link function were used for the statistical analyses of mating latency and other measurements, respectively. The genotypes of males and females and their interaction were included as fixed factors, because the full model has the lowest Akaike's information criterion score. Individual ID and the date of the experiment were included as random intercepts. To analyze the count data, mating latency was included as a log-linear offset to standardize the event numbers per unit time. For a post hoc test, *P* values adjusted with Tukey's method were calculated using the package *emmeans* version 1.4.7.

**Predator preference test.** Halfbeaks (*Nomorhamphus* cf. *ebrardti*), which were originally caught in Fotuno Fountain, were provided by the World's Medaka Aquarium, Nagoya Higashiyama Zoological Park. They were kept in 60 L glass tanks and fed *Artemia* nauplii, frozen red mites, and powdered food. A blue

polypropylene tank (546 mm width × 410 mm length × 332 mm height) (San box #73, Sanko, Tokyo, Japan), into which two 2-L plastic tanks (175 mm width × 120 mm length × 140 mm height) (Placase Mini, Nisso, Osaka, Japan) were inserted, was filled with 15 L of water at 26 °C (Supplementary Fig. 16a). The tank was illuminated by two white LED lights (LT-NLD10D-HA, Ohm Electronic), whose light spectra (Supplementary Fig. 17) and intensity (17.76 μmol m$^{-2}$ s$^{-1}$; 1193 Lux) were obtained using a light analyzer (LA-105, NK System). A wild-type *O. woworae* male (standard length±SD: 27.35 ± 0.99 mm, $n = 5$) was placed into one of the plastic tanks, and a *csf1*-KO male (standard length±SD: 26.99 ± 1.06 mm, $n = 5$) was put into the other. After a male halfbeak (standard length±SD: 47.25 ± 3.35 mm, $n = 13$) was positioned at the center of the swimming area, the behavior was recorded for 20 min by a web camera C920n (Logicool, Tokyo, Japan).

The positions of the anterior ends of each halfbeak were extracted using DeepLabCut v2.1.6.4[124,125] installed on Google Colaboratory (https://colab.research.google.com/). In brief, 30 images that were extracted from a recorded video and labeled manually were used for the network training with 75,500 iterations. Another video was analyzed using the trained network and then 30 putative outlier frames were extracted. The network was trained again after manual refinement of the marker positions in the outliers, followed by an additional three rounds of iterations (first round: 80,000 iterations; second round: 322,500 iterations; third round 173,500 iterations). The estimated positions with a low confidence prediction (likelihood < 0.9) or out of the *Nomorhamphus* swimming area were excluded. The trajectory of the anterior tip of each halfbeak was smoothed using the "TrajSmoothSG" function (with $p = 3$ and $n = 31$) in the R package *trajr* version 1.3.0[120] and was plotted using the "geom_path" function in the R package *ggplot2* (Supplementary Fig. 16b). A 20-mm length of the area was defined in the front of each plastic tank where the *Oryzias* fish as an "interaction zone" (Supplementary Fig. 16a), and the following values were measured: time spent in each zone, the number of entries into each zone, and latency to the first entry into each zone.

The time spent in each zone was statistically analyzed using LMMs by the "lmer" function in the R package *lmerTest* version 3.1–2. The number of entries and the latency to the first entry were analyzed by GLMMs using a Poisson distribution with a log link function and a gamma distribution with a log link function, respectively, by the "glmer" function in the R package *lme4* version 1.1-23. Each model included the genotype of the medaka as a fixed factor and the individual IDs of the medakas and halfbeaks as random intercepts.

**Reporting summary**. Further information on research design is available in the Nature Research Reporting Summary linked to this article.

## Data availability

All sequence reads are available from DDBJ: sequence reads used for de novo assembly of *O. celebensis* (DRA010635) and *O. woworae* (DRA011275), whole-genome re-sequences (DRA010665); RNA-seq of multiple tissues for gene annotation (DRA010666); differential expression analysis by RNA-seq (DRA010667); ddRAD-seq reads for linkage mapping (DRA010679). Assembled reference sequence of *O. celebensis* (BNCR01000001-BNCR01000594) and *O. woworae* (BOLG01000001-BOLG01001666) are also available from DDBJ. Gene annotations for the reference assembly of *O. celebensis* are available from a GitHub repository (https://github.com/satoshi-ansai/OryCel_1.0/). The Ensembl protein databases (Release 94, http://oct2018.archive.ensembl.org/) were used for gene annotation: *O. latipes* (ASM223467v1), *O. melastigma* (Om_v0.7.RACA) and *D. rerio* (GRCz11). The reference genome assemblies and their gene annotations in Ensembl Release 100 (http://apr2020.archive.ensembl.org/) were used for phylogenomic analysis: *O. latipes* (ASM223467v1), *O. sakaizumii* HNI (ASM223471v1), *O. latipes* HSOK (ASM223469v1), *O. sinensis* (ASM858656v1), *O. melastigma* (Om_v0.7.RACA), *O. javanicus* (OJAV_1.1), and *X. maculatus* (X_maculatus-5.0-male). Source data are provided with this paper.

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

## Acknowledgements

We thank the Ministry of Research, Technology, and Higher Education, Republic of Indonesia (RISTEKDIKTI) and the Faculty of Fisheries and Marine Science, Sam Ratulangi University, for the permission to conduct research in Sulawesi (research permit numbers 394/SIP/FRP/SM/XI/2014, 397/SIP/FRP/SM/XI/2014, and 106/SIP/FRP/E5/Dit. KI/IV/2018). We also thank the Kitano laboratory, Yamahira laboratory, Naruse laboratory, and Takeuchi laboratory members for discussion and technical assistance; Katie Peichel (University of Bern) and Mark Ravinet (The University of Nottingham) for their helpful comments on the manuscript; Thomas von Rintelen (Museum für Naturkunde) for providing the map of Sulawesi; NBRP medaka for providing *O. celebensis* (Ujung pandang); R. Tanaka (Nagoya Higashiyama Zoological Park) for providing halfbeaks; T. Sue (Picta Ltd.) for the assistance in rearing experimental fish; T. Yamazaki, I. Hara, J. Sakamoto, Y. Kamei (NIBB) and S. Kondo (Ryukoku University) for their technical assistance; S. Kanda (University of Tokyo) for providing a behavioral recording system with Raspberry Pi; S. Higashijima (NIBB) for providing plasmids. This work was supported by JSPS KAKENHI Grant Number 18K14769 to S.A., 16K14792 to J.K., 26291093 to K.Y., 19K16232 to S.F., and 16H06279 (PAGS) to K.Y. and MEXT KAKENHI (16H06279) to A.T., the JSPS Research Fellowship for Young Scientists (16J05534) to S.A., NIBB Collaborative Research Program (17-313) to J.K., the Collaborative Research of Tropical Biosphere Research Center, University of the Ryukyus to J.K. and NIG-JOINT (20A2018, 20A2019, and 5B2020) to S.A. We thank the Spectrography and Bioimaging Facility and the Functional Genomics Facility, NIBB Core Facilities, for technical assistance. Computations were partially performed on the Biological Information Analysis System provided by the Data Integration and Analysis Facility, NIBB, and the NIG supercomputer at ROIS NIG.

## Author contributions

S.A., K.Y., and J.K. conceived and designed the research. S.A., K.M., S.F., D.F.M., A.J.N., A.T., K.Y., and J.K. performed the experiments. S.A. analyzed the data. D.F.M., B.K.A.S., K.W.A.M., R.K.H., and K.N. contributed materials/analysis tools. S.A. and J.K. wrote the manuscript with input from the other authors.

## Competing interests

The authors declare no competing interests.
