## [Peer Review File · Nature Communications]

Reviewer #1 (Remarks to the Author):

The manuscript "Fitness effects of a gene for sexual dimorphism revealed by genome editing in a Sulawesi fish" by Ansai et al. is basically four manuscripts in one. The authors have first generated a high quality, chromosome resolution reference genome for *Oryzias celebensis* a species of Sulawesi medakas. Then they sequenced whole genomes of 17 other species of Sulawesi medakas and resolved the phylogenetic relationship of this clade. Then they performed a QTL analysis for a sex-specific color trait using two closely related species with different fin coloration. Using a combination of transcriptome analysis and very elegant genome editing experiments the authors provide strong evidence for the gene *csf1* to be underlying the sex-specific color trait. Finally, they use the mutants they generated to test the role of the fin coloration on mate preference and predator avoidance. All these parts are beautifully executed and of extremely high quality. This is the best paper I have read since Colosimo 2005 and sets a new standard in the field. Every part would be an important paper by itself, but in combination it punches way above all other current work in fish evo geno. Simply amazing. Congrats on the authors for such beautiful and impressive work.

I only have some minor points that are basically just suggestions to improve the style of the paper. If the authors decide to ignore these completely, I am totally fine with it and my opinion remains unchanged. This paper can be published as it is and I can't wait to see it published.

Minor points:

Title is a little uninformative. I personally would mention the gene name and the gene has been revealed by much more than just genome editing. But it's just personal style or preference.

Line 147: The QTL for standard length could go into the supplement as it is a bit distracting.

Line 167: I would like to see all five candidate genes mentioned in the main text

Line 221: I would add here a sentence that the authors screened the sequence surrounding the gene for androgen receptor binding sites but no obvious alterations were found and future studies will be needed to identify the precise molecular basis for this (similar to what they mention in the discussion).

Reviewer #2 (Remarks to the Author):

The manuscript "Fitness effects of a gene for sexual dimorphism revealed by genome editing in a Sulawesi fish" by Ansai et al. brings together a wide range of cutting edge techniques (both molecular and behavioral) to address an important outstanding question in evolutionary biology – How do sexually dimorphic traits involved in mate choice arise at the genomic level? The work presented here has direct relevance to a broad range of biological disciplines including evolution, genetics, behavioral ecology, and genomics. Additionally, sexually dimorphic traits have a broad interest to the public at large and I feel this work will make an excellent contribution to Nature Communications.

The scientific rigor of this manuscript is exceptional, each step is appropriately tested, data and conclusions are presented in a clear and concise manner. All in all, this body of work is an excellent example of how future work integrating genomics/behavior/genetics/evolution should be conducted and I commend the authors on their accomplishment. As such I have only very minor comments to contribute.

Minor comments:

L.218 – When presenting the increase in expression of *csf1* in females with MT treatment (3.63X) I had to go back and find how much higher males were than females (3.52X). That these are nearly identical is exceptionally strong support for your conclusions that *csf1* sex

differences are due to testosterone. Perhaps you could remind the reader here of the male:female differences so they could easily see how similar these are.

L. 345 – remove “was used”.

L. 498 – “... in at least in one...” to “... in at least one...”

L. 703 – Was there more than one halfbeak originally caught?

Figure 1 – Perhaps you could mark the focal and reference species next to their photos?

Supplementary Figure 4 – Could you add the *csf1* gene to this figure? Just looking at this figure currently it is difficult to compare other candidate genes to *csf1*. The CRISPR/Cas experiments clearly resolve that *csf1* is the most likely candidate (which can be mentioned in the figure legend).

Supplementary Figure 10 – I don't see “Lines connect the same female individuals.” (L. 1196-1197). Perhaps these were lost?

As a side note- I think the authors could make more of their results of female preference for males with and without red ornamentation. While indeed wildtype (*csf1*^{+/+}) females had a positive preference for red males, the *csf1*^{-/-} females did not show a preference for red male ornamentation (as presented in Figure 4). I am extremely excited by this result – there are many models for the evolution of female preferences, but it is often very difficult to distinguish among these models with the data available. This is the first example I know of that the gene for a sexually selected trait could be the same as the gene for female preference of that trait (a key mechanism for the Fisherian Runaway model of female preference). While it may be out of the scope of this manuscript to really dig into this line of research, I hope to see more from the authors on this subject in the future as it will help resolve several long standing questions in the field of sexual selection.

All in all- I am extremely excited by the work presented here and look forward to reading more from this group on this fantastic line of research.

- Ben Sandkam

Reviewer #3 (Remarks to the Author):

Authors provided a reference genome sequence of *Oryzias celebensis* for a representative of diverse Slawesian medaka species, and then detangled the genetic basis underlying the expression of male-specific red coloration in *Oryzias woworae*. The authors identified *csf1* as a primary causative gene for the red coloration on pectoral fins in the males. Authors also indicated the *csf1*-mediated red coloration led to reproduction advantage and predator avoidance.

I think successive functional genomic analyses including genome re-sequencing, QTL analysis, RNA-seq, and genome editing provided in the study are excellent. I found the paper to be interesting and of importance for our understanding of the evolution of diverse sexually-selected traits at a genetic level. However, I believe the manuscript has several points should be further addressed especially on (1) the mechanism underlying the *csf1* expression change, (2) fitness effects of *csf1*-mediated red coloration, and (3) plausible ecological background for the evolution of red pectoral fin in *O. woworae* male. I provide a list of detailed comments:

(1) Authors concluded that the cis-regulatory mutation and subsequent changes in androgen-dependent regulation of transcription caused the *csf1* expression change. This is a most likely scenario but a bit more evidence should be presented in the manuscript. My questions are:
- Can the allele-specific expression change in *csf1* be observed in F2 females by the administration of methyltestosterone?

- Are there some hormone response elements (androgen response elements) on the upstream region of *csf1*?
- The difference in *csf1* expression between *O. woworae* and *O. asinua* is age-dependent? (Does the expression level of *csf1* in *O. woworae* relative to *O. asinua* vary with the development/growing stage?)
- The changes in *csf1* expression is not observed in *O. asinua* by the MT administration?

(2) I have several concerns about the procedure of the behavioral experiments.

- Is there possibility that females reared with red males have a preference for red males? Generally, familiarity can have an influence in mate choice criteria.
- I think the light condition (light intensities and irradiance spectra) in the mating test and predator preference test should be provided. How was the relevance to the natural habitat conditions?
- The effect of *csf1* KO on predator preference was male-specific? Is there any effect in females?

(3) I don't think the authors have put enough information on or thought into the ecological background of the evolution of red pectoral fins in *O. woworae*. Why the exaggerated coloration has been evolved in this species? The female preference for the red coloration in males is unique in *O. woworae* or that is shared in multiple species? I would like authors to provide descriptions on these in Introduction or Discussion section.

Also, I provide other specific comments/questions below:

Introduction:

L. 80~ - 88: Can the interspecific crossing among Sulawesian medaka species be observed in the natural setting?

Results:

L. 167~: Information on the other four candidate genes (*magi1a*, *pcbp4*, *ren* and *smapb*) should be also provided briefly.

Does the MT-induced upregulation of *csf1* in *O. woworae* female cause the expression of red coloration?

How was the difference in predator attraction between wild-type and *csf1*-KO females?

Methods:

L. 536~: The description of method for binary evaluation of color traits was ambiguous, so please clarify it. Species-specific colorations could be discretely categorized? Any intermediate traits were not found?

Figure 3i: Why was the ratio of the *O. woworae* allele/the *O. celebensis* allele higher even in the genome DNA (the value should be zero, I think)?

Figure 4: Please provide the reflectance spectral data (especially S1. Red [605 ~ 700 nm]) on the wild-type and *csf1*-KO males, in addition to the representative images?

Response: Here are our point-by-point responses to the reviewers' comments. We highlighted all changes in red letters in the main text.

Reviewer #1:

*The manuscript "Fitness effects of a gene for sexual dimorphism revealed by genome editing in a Sulawesian fish" by Ansai et al. is basically four manuscripts in one. The authors have first generated a high quality, chromosome resolution reference genome for *Oryzias celebensis* a species of Sulawesian medakas. Then they sequenced whole genomes of 17 other species of Sulawesian medakas and resolved the phylogenetic relationship of this clade. Then they performed a QTL analysis for a sex-specific color trait using two closely related species with different fin coloration. Using a combination of transcriptome analysis and very elegant genome editing experiments the authors provide strong evidence for the gene *csf1* to be underlying the sex-specific color trait. Finally, they use the mutants they generated to test the role of the fin coloration on mate preference and predator avoidance. All these parts are beautifully executed and of extremely high quality. This is the best paper I have read since Colosimo 2005 and sets a new standard in the field. Every part would be an important paper by itself, but in combination it punches way above all other current work in fish evo geno. Simply amazing. Congrats on the authors for such beautiful and impressive work.*

I only have some minor points that are basically just suggestions to improve the style of the paper. If the authors decide to ignore these completely, I am totally fine with it and my opinion remains unchanged. This paper can be published as it is and I can't wait to see it published.

Response: Thank you very much for finding our work "amazing".

Title is a little uninformative. I personally would mention the gene name and the gene has been revealed by much more than just genome editing. But it's just personal style or preference.

Response: The title was changed to "Genome editing reveals fitness effects of *Colony stimulating factor 1*, a gene responsible for diversification of sexual dichromatism in Sulawesian fishes" to be more informative.

Line 147: The QTL for standard length could go into the supplement as it is a bit distracting.

Response: We think that some readers would be interested in whether other traits are linked to color. We therefore left the QTL for the standard length in the main Figure 2, from which readers can easily compare QTL for different traits. If the editor considers that movement of this data to the

supplement is necessary, we will follow the editor's discretion.

Line 167: I would like to see all five candidate genes mentioned in the main text

Response: All five genes are mentioned in the main text (L164) and the legend of Supplementary Figure 4 as suggested.

Line 221: I would add here a sentence that the authors screened the sequence surrounding the gene for androgen receptor binding sites but no obvious alterations were found and future studies will be needed to identify the precise molecular basis for this (similar to what they mention in the discussion).

Response: We investigated the positions of a putative androgen receptor binding motif at the *csfl* locus in the two species (*O. woworae* and *O. celebensis*). We found that they are overall similar, but there are some differences. This result is summarized in Supplementary Figure 10 and described in the main text (L234-236): “Comparison of putative androgen response elements (AREs) near *csfl* between *O. woworae* and *O. celebensis* revealed several variations in AREs in the non-coding regions (Supplementary Figure 14).”.

Reviewer #2:

The manuscript “Fitness effects of a gene for sexual dimorphism revealed by genome editing in a Sulawesian fish” by Ansai et al. brings together a wide range of cutting edge techniques (both molecular and behavioral) to address an important outstanding question in evolutionary biology – How do sexually dimorphic traits involved in mate choice arise at the genomic level? The work presented here has direct relevance to a broad range of biological disciplines including evolution, genetics, behavioral ecology, and genomics. Additionally, sexually dimorphic traits have a broad interest to the public at large and I feel this work will make an excellent contribution to Nature Communications.

The scientific rigor of this manuscript is exceptional, each step is appropriately tested, data and conclusions are presented in a clear and concise manner. All in all, this body of work is an excellent example of how future work integrating genomics/behavior/genetics/evolution should be conducted and I commend the authors on their accomplishment. As such I have only very minor comments to contribute.

Response: Thank you very much for finding our paper would “*make an excellent contribution to*

Nature Communications".

Minor comments:

L.218 – When presenting the increase in expression of *csf1* in females with MT treatment (3.63X) I had to go back and find how much higher males were than females (3.52X). That these are nearly identical is exceptionally strong support for your conclusions that *csf1* sex differences are due to testosterone. Perhaps you could remind the reader here of the male:female differences so they could easily see how similar these are.

Response: Thank you very much for excellent suggestion. As suggested, we mention this in the revised manuscript (L222-226): "Administration of a synthetic androgen methyltestosterone (MT) in *O. woworae* females increased the expression level of *csf1* by 3.63 times (Welch's *t*-test, $t_{12,233} = -5.4148$, $P = 0.00015$; Figure 3j; Supplementary Figure 10), which is nearly identical to the difference between *O. woworae* males and females (3.52 times higher in males; Figure 3d)."

L. 345 – remove "was used".

Response: Corrected as suggested (L384: "was used" after "heterozygosity" was removed).

L. 498 – "... in at least in one..." to "... in at least one..."

Response: Corrected as suggested (L537).

L. 703 – Was there more than one halfbeak originally caught?

Response: Yes, we caught more than one halfbeak. We changed "A halfbeak..." to "Halfbeaks" (L790).

Figure 1 – Perhaps you could mark the focal and reference species next to their photos?

Response: We marked these species using arrow heads in Figure 1 and explained it in the figure legend (L1166-1167): "Open and closed arrowheads indicate the reference and focal species in this study, respectively."

Supplementary Figure 4 – Could you add the *csf1* gene to this figure? Just looking at this figure currently it is difficult to compare other candidate genes to *csf1*. The CRISPR/Cas experiments

clearly resolve that csf1 is the most likely candidate (which can be mentioned in the figure legend).

Response: We added the data on the expression level of the *csf1* to Supplementary Figure 4 as suggested.

Supplementary Figure 10 – I don't see "Lines connect the same female individuals." (L. 1196-1197). Perhaps these were lost?

Response: Apologies for the error. This sentence was erroneously left from a much earlier version of the paper. It is now removed.

As a side note- I think the authors could make more of their results of female preference for males with and without red ornamentation. While indeed wildtype (csf1 +/+) females had a positive preference for red males, the csf1-/- females did not show a preference for red male ornamentation (as presented in Figure 4). I am extremely excited by this result – there are many models for the evolution of female preferences, but it is often very difficult to distinguish among these models with the data available. This is the first example I know of that the gene for a sexually selected trait could be the same as the gene for female preference of that trait (a key mechanism for the Fisherian Runaway model of female preference). While it may be out of the scope of this manuscript to really dig into this line of research, I hope to see more from the authors on this subject in the future as it will help resolve several long standing questions in the field of sexual selection.

Response: We provided an additional paragraph in the discussion section to discuss the possibility that the *csf1* gene controls the female mate preferences (L344-352): “Although female mate preference is one of the key factors for the evolution of male mating signals^{1,2,56}, we currently know little about the molecular genetic basis for female mate preference. Our data showed a trend that *csf1* knock-out females have less preference for red males compared to the wild-type females (Figure 4e). This suggests that *csf1* may control not only male mating signals but also female mate preference. Theoretical studies indicate that linkage of male mating signals and female mate preference can promote the evolution of exaggerated male ornamental traits^{20,57,58}. As CSF-1 deficiency in neuronal cells is shown to alter neuronal functions and the social preference in mice⁵⁹, it is possible that *csf1* has a pleiotropic effect on both pigment cell and brain development.”

Comments by Reviewer #3:

Authors provided a reference genome sequence of Oryzias celebensis for a representative of diverse Slawesian medaka species, and then detangled the genetic basis underlying the expression of

male-specific red coloration in Oryzias woworae. The authors identified csfl as a primary causative gene for the red coloration on pectoral fins in the males. Authors also indicated the csfl-mediated red coloration led to reproduction advantage and predator avoidance.

I think successive functional genomic analyses including genome re-sequencing, QTL analysis, RNA-seq, and genome editing provided in the study are excellent. I found the paper to be interesting and of importance for our understanding of the evolution of diverse sexually-selected traits at a genetic level. However, I believe the manuscript has several points should be further addressed especially on (1) the mechanism underlying the csfl expression change, (2) fitness effects of csfl-mediated red coloration, and (3) plausible ecological background for the evolution of red pectoral fin in O. woworae male. I provide a list of detailed comments:

Response: Thank you very much for finding “*the paper to be interesting and of importance for our understanding of the evolution of diverse sexually-selected traits at a genetic level*”.

*(1) Authors concluded that the cis-regulatory mutation and subsequent changes in androgen-dependent regulation of transcription caused the csfl expression change. This is a most likely scenario but a bit more evidence should be presented in the manuscript. My questions are:
- Can the allele-specific expression change in csfl be observed in F2 females by the administration of methyltestosterone?*

Response: This is a very interesting idea. However, all F₂ fish were already sacrificed for the QTL analysis and it will take another year for making a new hybrid cross. Instead, we conducted additional experiments using pure species crosses (*O. woworae* females and *O. asinua* females). We found that androgen treatment increased the *csfl* expression in the pectoral fins of *O. woworae* females but not in those of *O. asinua* females (*O. asinua* does not have red pectoral fins in either sex). This supports our claim that changes in androgen-dependent regulation of transcription is important for the divergence in red coloration. We added these results in L220-230: “As androgen-dependent gene regulation often mediates male-biased expression of autosomal genes in vertebrates^{22,23,25}, we examined the effects of androgen on *csfl* expression and red coloration of the pectoral fins. Administration of a synthetic androgen methyltestosterone (MT) in *O. woworae* females increased the expression level of *csfl* by 3.63 times (Welch’s *t*-test, $t_{12,233} = -5.4148$, $P = 0.00015$; Figure 3j; Supplementary Figure 10), which is nearly identical to the difference between *O. woworae* males and females (3.52 times higher in males; Figure 3d). The females administered with MT also had a significantly larger red pigmented area in the pectoral fin than the control females (Supplementary Figure 11). On the other hand, MT administration in females of *O. asinua*, a species

that does not show red coloration in either sex, did not increase the *csfl* expression in the pectoral fins (Supplementary Figure 12).”

"

- Are there some hormone response elements (androgen response elements) on the upstream region of csfl?

Response: We compared androgen receptor binding motifs in the upstream region of *csfl* between two genome assemblies (*O. woworae* and *O. celebensis*) and found some differences. The data were described in the main text (L234-236) and Supplementary Figure 10: “Comparison of putative androgen response elements (AREs) near *csfl* between *O. woworae* and *O. celebensis* revealed several variations in AREs in the non-coding regions (Supplementary Figure 14).”

- The difference in csfl expression between O. woworae and O. asinua is age-dependent ? (Does the expression level of csfl in O. woworae relative to O. asinua vary with the development/growing stage?)

Response: We do not have any data on the age-dependency of *csfl* expression. Instead, we analyzed the expression levels in 3-4 month-old fish with varied body size and found larger fish have significantly higher *csfl* expression levels. This is mentioned in the main text (L230-232) and Supplementary Figure 13: “Additionally, we found that larger fish have higher expression levels of *csfl* in *O. woworae* (Supplementary Figure 13), suggesting that *csfl* expression may change during growth.”

- The changes in csfl expression is not observed in O. asinua by the MT administration?

Response: We showed that the MT treatment did not affect the *csfl* expression levels in the pectoral fins of *O. asinua* females (Supplementary Figure 12): “On the other hand, MT administration in females of *O. asinua*, a species that does not show red coloration in either sex, did not increase the *csfl* expression in the pectoral fins (Supplementary Figure 12)”. (L228-230)

(2) I have several concerns about the procedure of the behavioral experiments.

- Is there possibility that females reared with red males have a preference for red males? Generally, familiarity can have an influence in mate choice criteria.

Response: To avoid the possibility that familiarity with males of particular genotypes affects female preference, all tested females were reared with both red and non-red males. Therefore, the females

were familiar with both males. We described this point more clearly in the method section (L752-757): “To exclude the possibility that females choose familiar males with particular genotypes, individuals with different genotypes were reared in a single 3-L tank: a single rearing tank contained one wild-type (*csf1*^{+/+}) male, one wild-type (*csf1*^{+/+}) female, one homozygous mutant (*csf1*^{Δ11/Δ11}) male and one homozygous mutant (*csf1*^{Δ11/Δ11}) female. Therefore, the tested females were familiar with both wild-type and mutant males.”.

- I think the light condition (light intensities and irradiance spectra) in the mating test and predator preference test should be provided. How was the relevance to the natural habitat conditions?

Response: We described the light condition in the behavioral experiments in the method section and Supplementary Figure S17. The light sources covered the wavelength of 400-750 nm, although we admit that their spectra might differ from those of the natural habitat.

*- The effect of *csf1* KO on predator preference was male-specific? Is there any effect in females?*

Response: Thank you for the excellent suggestion. However, we do not have any data on this, and currently there are no halfbeak fish available for doing this experiment. We hope that the reviewers and editors think that our revised paper is interesting enough without this data.

*(3) I don't think the authors have put enough information on or thought into the ecological background of the evolution of red pectoral fins in *O. woworae*. Why the exaggerated coloration has been evolved in this species? The female preference for the red coloration in males is unique in *O. woworae* or that is shared in multiple species? I would like authors to provide descriptions on these in Introduction or Discussion section.*

Response: We made a new paragraph in Discussion and described possible ecological factors that influenced the evolution of the red coloration. We do not know female preference in other species and mentioned that such analysis is necessary in the future (L334-343): “Several environmental factors of the habitats may contribute to the evolution of the red coloration in *O. woworae*. The natural habitat of the *O. woworae* population used in this study is dominated by light with shorter-wavelengths³⁸. Such environments generally create blue background and make males with red colorations more conspicuous to conspecific females⁵¹. Such sensory drive may promote the evolution of red coloration in *O. woworae*. Behavioral analysis of female mate preference and predation experiments under manipulated light conditions, which have been successfully conducted in other fishes^{54,55}, will lead to a better understanding of the environment-dependent fitness effects of

the *csfl* gene in *O. woworae*. Analysis of female preference for red coloration in other species that do not show red coloration is also necessary.”

Also, I provide other specific comments/questions below:

Introduction:

L. 80~ – 88: Can the interspecific crossing among Sulawesian medaka species be observed in the natural setting?

Response: We mentioned this in Discussion (L284-288): “Furthermore, several Sulawesian medaka species are sympatric and have undergone interspecies hybridization, making them a good model system for studying the mechanisms of speciation and hybridization³⁹, although *O. celebensis* and *O. woworae* used in this study are allopatric with no hybrids observed in the wild.”

Results:

L. 167~: Information on the other four candidate genes (magi1a, pcbp4, ren and smapb) should be also provided briefly.

Response: We mentioned this point in the figure legend of Supplementary Figure S4: “none of these genes except *csfl* are known to be expressed or function in pigment cells.”

Does the MT-induced upregulation of csfl in O. woworae female cause the expression of red coloration?

Response: Yes. We added the data in Supplementary Figure S11.

How was the difference in predator attraction between wild-type and csfl-KO females?

Response: As described in Response to the major comment (2), we do not have any data on this, and there are no halfbeak fish available for doing this experiment.

Methods:

L. 536~: The description of method for binary evaluation of color traits was ambiguous, so please clarify it. Species-specific colorations could be discretely categorized? Any intermediate traits were not found?

Response: Red pectoral fin and black stripes in the tail fin were inherited as discrete traits while blue

in the lateral body was inherited as a continuous trait. Importantly, QTL analysis using reflectance spectra for the red and blue traits was qualitatively similar to that with the binary analysis. To clarify how we categorized the binary traits, we explained the details in the methods (L576-579): “For the binary analysis of coloration traits, F₂ fish were classified according to the presence or absence of pigment cells (red pigment cells in the dorsal side of the pectoral fin and iridophores in the lateral body) and black stripes of melanophores in the tail fin.”

Figure 3i: Why was the ratio of the O. woworae allele/the O. celebensis allele higher even in the genome DNA (the value should be zero, I think)?

Response: This may be due to the differences in the amplification efficiencies of PCR and/or cycle sequencing between the two alleles. However, the ratios in cDNA were far higher than in genomic DNA (as shown in Figure 3i).

Figure 4: Please provide the reflectance spectral data (especially Sl. Red [605 – 700 nm]) on the wild-type and csfl-KO males, in addition to the representative images?

Response: We measured the reflectance in the *csfl* wild-type and knock-out fish and showed the data in Supplementary Figure 7. Accordingly, we made a new section in the Methods (L733-740).

We would like to thank all reviewers and editors for taking the time to review our paper and making excellent feedbacks. We hope that the revised paper is now suitable for publication in *Nature Communications*.

Reviewer #2 (Remarks to the Author):

I have now read the updated manuscript and find that the authors have addressed all of my concerns. I once again commend the authors on an impressive study and look forward to reading their future work.

Sincerely,
Ben Sandkam, PhD
Assistant Professor
Cornell University

Reviewer #3 (Remarks to the Author):

I'd like to thank the authors for addressing my previous concerns and suggestions. I am happy with the revision. I believe this work will be of significance to the ecological evolutionary genomics and the related fields. Only a few minor comments are listed below.

Line 116 and Figure 1: Why are the numbers of genes shown in the main text (10,810) different from those in the capture of Figure 1 (10,174)?

Line 118: "[BI]" to "[CI]"?

Line 144: "LG15" to "LG15 and LG12_20_13"

Some of the figure numbers referred to in the main text appear to be wrong. You could double-check them?

ex)

Line 130: "Figure 2b" to "Figure 2a"

Line 180 and 181 "Figure e, g" to "Figure f, g"

Figure 3h. Is the expression shown here also that of the pectoral fin?

Supplementary Figure 3f: The position of the nearest marker (chr12_20_13: 31,532,462) is correct? (I saw that was of the black stripe.)

Supplementary Figure 5: For clarity, could you highlight the *Oryzias csf1* genes and also mark the *csf1* of *O. woworae* in the phylogenetic tree?

Response: Here are our point-by-point responses to the reviewers' comments. We highlighted all changes in red letters in the main text.

Reviewer #2 (Remarks to the Author):

I have now read the updated manuscript and find that the authors have addressed all of my concerns. I once again commend the authors on an impressive study and look forward to reading their future work.

Response: Thank you very much for finding our work impressive.

Reviewer #3 (Remarks to the Author):

I'd like to thank the authors for addressing my previous concerns and suggestions. I am happy with the revision. I believe this work will be of significance to the ecological evolutionary genomics and the related fields. Only a few minor comments are listed below.

Line 116 and Figure 1: Why are the numbers of genes shown in the main text (10,810) different from those in the capture of Figure 1 (10,174)?

Response: We apologize for the error. The number was wrong, and we corrected it to 10,174 (L111). Thank you for pointing this out.

Line 118: “[BI]” to “[CI]”?

Response: Thank you again for pointing this out. Corrected as suggested (L113: “BI” was changed to “CI”).

Line 144: “LG15” to “LG15 and LG12_20_13”

Response: The another QTL was not mentioned in the main text in the original manuscript, and we have corrected as suggested (L139: “LG15” was changed to “LG12_20_13 and LG15”).

Some of the figure numbers referred to in the main text appear to be wrong. You could double-check them?

ex)

Line 130: “Figure 2b” to “Figure 2a”

Line 180 and 181 “Figure e, g” to “Figure f, g”

Response: Apologies. The following figure numbers were corrected.

L125: “Figure 2b” to “Figure 2a”

L175 and L176: “Figure 3e, g” to “Figure 3f, g”

Figure 3h. Is the expression shown here also that of the pectoral fin?

Response: Yes. To clarify it, we modified the legend of Figure 3h as follows: “Expression levels of *csf1* in the pectoral fin of the F₂ family (*n* = 8 for each)” (L1214-1215).

Supplementary Figure 3f: The position of the nearest marker (chr12_20_13: 31,532,462) is correct? (I saw that was of the black stripe.)

Response: Apologies. The nearest marker name for the standard length was not correct in Supplementary Figure 3f. We changed it to the correct name “chr12_20_13:66,117,828”.

*Supplementary Figure 5: For clarity, could you highlight the *Oryzias csf1* genes and also mark the *csf1* of *O. woworae* in the phylogenetic tree?*

Response: The *csf1* genes of *Oryzias* fishes and the focal species (*O. woworae csf1* gene) were highlighted in a blue box and red letters, respectively.

We would like to thank again all reviewers and editors for taking the time to review our paper.